# Study on the sustainability of ancient canal towns on the basis of the topological structure analysis of streets and lanes: A case study of the Xixing ancient town

**Weilu Lv[1,2], Ning Wang[3]\*, Yun Huang[4]**

1 School of Art and Archaeology, Zhejiang University City College, Hangzhou, China, 2 School of Architecture, Tsinghua University, Beijing, China, 3 School of Spatial Planning and Design, Zhejiang University City College, Hangzhou, China, 4 Zhijiang College of Zhejiang University of Technology, Shaoxing, China

\* wangning@zucc.edu.cn

**Data Availability Statement:** All relevant data are within the paper and its Supporting Information files.

## Abstract

In China, rapid urbanization and the decline of the canal's shipping function have directly led to the decline of the space and style of ancient towns with canals. The ancient town of Xixing, known to be the first canal town, is located west of the Zhedong canal section of the Grand Canal. It used to be the most active transportation hub and trade transit point in the east Zhejiang Province. However, the canal is now facing problems in protection and renewal, which require urgent attention. For example, the streets of the ancient town are separated from the water system space, the self-organizing and development ability of the ancient town has weakened, and the spatial vitality is scarce. Firstly, by comparing and analyzing whether the canal is used as a road network under two different spatial scales, the area with the highest degree of spatial integration is obtained before judging whether the canal has an important influence and control on the street spaces of the ancient town. Secondly, as the canal is no longer used as a transportation network nowadays, the internal spatial structure characteristics of streets and alleys are analyzed. Finally, the topological structure analysis is carried out in the 3 km area from the border of the planning control area, and the integrated development of the ancient town space and urban transportation network is then discussed. Further, the vitality renewal strategies of the ancient town under different spatial scales are put forward to provide the empirical basis and decision-making reference for the better sustainable development of ancient towns with canals. According to research, once the ancient town is considered on a larger urban scale, the areas with the best integration and accessibility, like the water-adjoined spaces change to intersection spaces and urban road network areas. Consequently, in order to realize the sustainable development of Xixing ancient town, the city system must overcome its shortfalls in operation. The methods and conclusions in the paper can effectively contribute to the development of the ancient town. The research significantly helps to improve the average land sharing, commercial distribution, and public facilities layout in the practice of protection and

**Funding:** The funders had no role in study design, data collection and analysis, decision to publish, or preparation of the manuscript. This research was funded by Zhejiang Philosophy and Social Science Fund 2019 (Grant Number: 19NDQN353YB) and Humanities and Social Sciences Youth Foundation, Ministry of Education of the People's Republic of China (Grant Number: 22YJCZH161).

**Competing interests:** The authors have declared that no competing interests exist.

development in ancient towns and provides an empirical basis and decision-making references for relevant management departments.

## 1. Introduction

Unlike natural drainage systems, canals, as artificially excavated waterways, are human pioneers in utilizing and transforming nature and promoting social development. They have multiple meanings and comprehensive economic, political, and cultural values. As the earliest and largest canal in the world, the Grand Canal is a great project on the eastern plain of China. It has been excavated for more than 2,500 years and has a total length of 1,797 km. In 2014, the 38th World Heritage Conference declared the Grand Canal as the 46th World Heritage Site in China [1]. Along the Grand Canal, through continuous self-evolution and change, many famous cultural cities and ancient towns of different scales and compositions have emerged, which carries the life, production, and cultural memory of the canal residents. However, due to the rapid process of urbanization and the decline of canal shipping functions, the original production and lifestyle of the ancient towns along the canal have changed in characteristics, and as a result, such ancient towns with canals lost their vitality through time.

The Xixing ancient town, now located in the Binjiang district of the Hangzhou City of Zhejiang Province, marks the beginning of the Zhedong canal section of the Grand Canal. More than 1700 years of evolution have resulted in a strip-like urban formation, "one water stripe running through the city" [2]. The ancient town has a road grid in the shape of a 'fish bone', where in the Xixing street and Guanhe road in the East-West direction form the main frame, with narrow lanes in the North-South direction. The streets (roads) in the ancient town are divided into two categories. Xixing Street, parallel to the canal, is distributed along the east-west direction and is about 2–3 m wide. The north to south lanes, perpendicular to the canal, are mostly 1 m wide and are marked by shops on the sides of the street. Small piers are set up every 20 m in the residential areas adjacent to the canal and are directly linked to Xixing street. The two banks of the canal are well connected, which facilitates three-dimensional transportation of water and land connection (see Fig 1).

As the first ancient town with canals in eastern Zhejiang, Xixing has long been an important distribution hub for goods. It is the most active transportation inter-spot in eastern Zhejiang, and goods are transported to all sides of the Grand Canal. Since 1990, Binjiang district, where the town is located, has evolved as the first national high-tech industrial development zone in China. In 2006, the ancient town was officially approved to be listed in the Planning for the Protection of Historical Blocks of Xixing Ancient Street in Hangzhou by the Hangzhou Municipal People's government. It became one of the top ten historical protection blocks in Hangzhou [3].

Comparing the satellite images from 1969 and 2021 (see Fig 2), we can see that the basic structure of the ancient town was preserved in the process of urbanization, but its usage attributes changed substantially. On the one hand, in 1977, the Beitang river (about 35 m wide), which was crucial for irrigation and shipping, was excavated to the north of Xixing ancient town. The original canal in the ancient town retained only some functions providing water and aesthetics [4]. On the other hand, in the then agricultural society, Xixing ancient town served as a commercial and trade distribution center, playing an important role in organization and communication. However, it became a historical and cultural patch embedded in modern cities as a well-protected object. Set in the new historical period, this paper proposes adaptive protection and renewal strategies to stimulate the spatial vitality of ancient towns and to better integrate them into urban development.

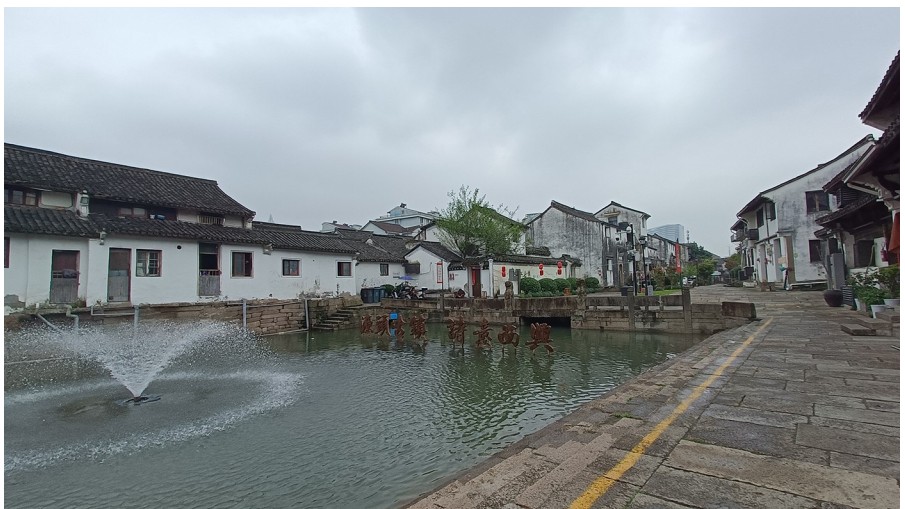

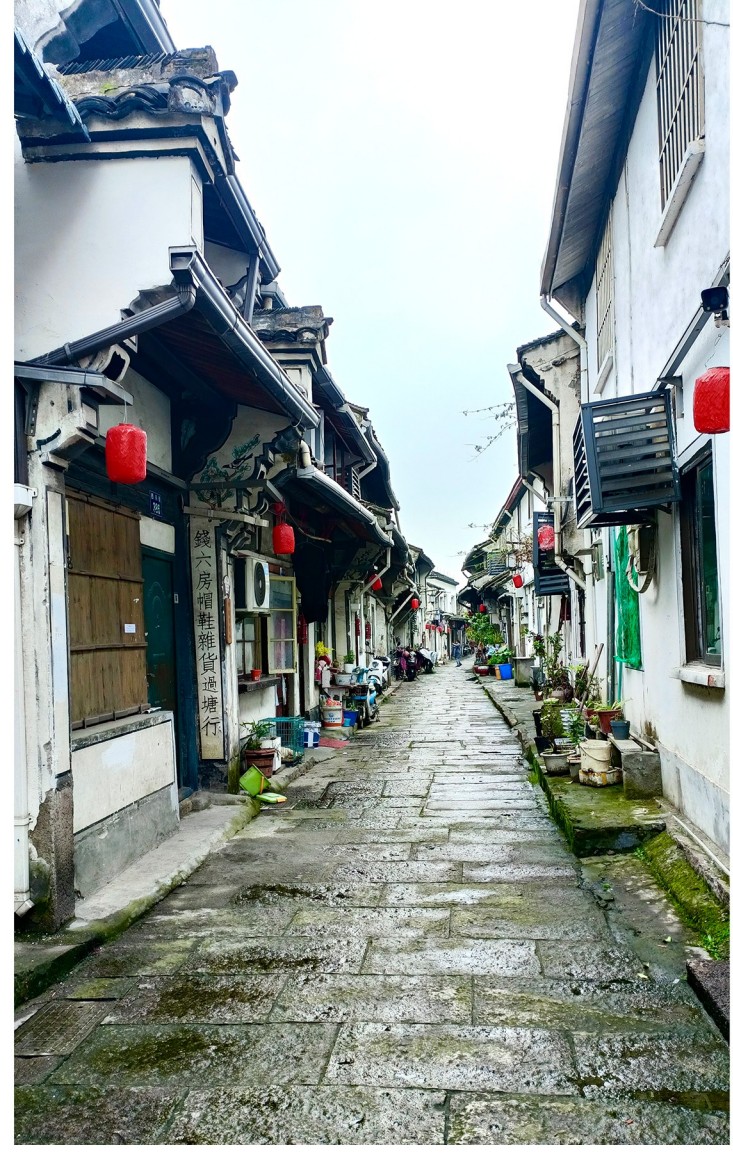

**Fig 1. Xixing ancient town present situation.** (a)Xixing townscape along the Zhedong Canal; (b) Xixing old street (Photoed by the authors).

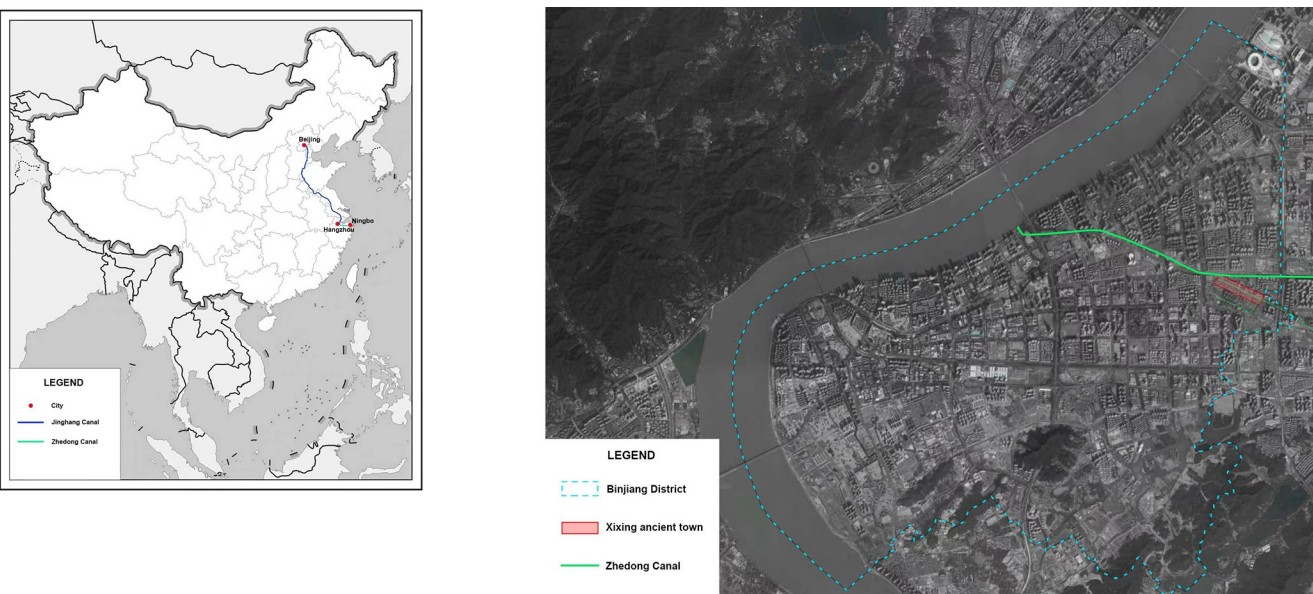

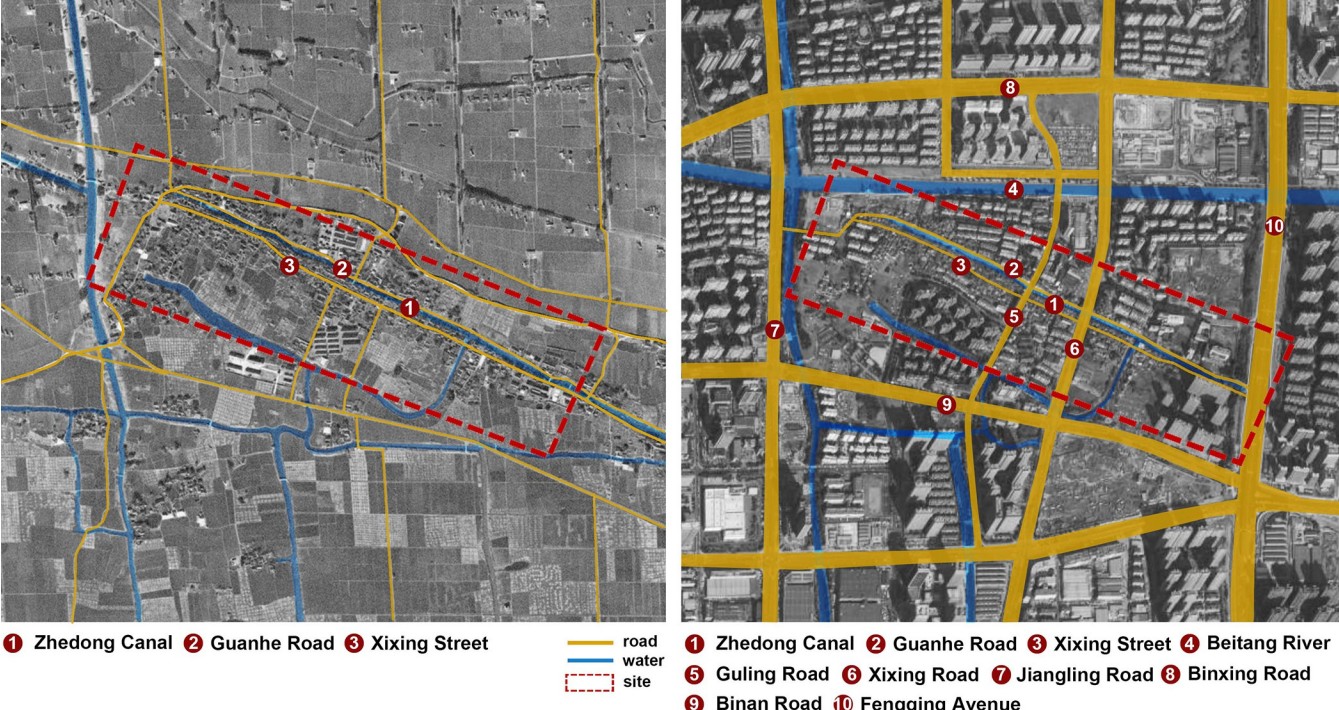

① Zhedong Canal ② Guanhe Road ③ Xixing Street

road
water
site

① Zhedong Canal ② Guanhe Road ③ Xixing Street ④ Beitang River
⑤ Guling Road ⑥ Xixing Road ⑦ Jiangling Road ⑧ Binxing Road
⑨ Binan Road ⑩ Fengqing Avenue

**Fig 2.** (a) The legend of canal and city. (b) The location of Xixing ancient town. (3) Comparison of Satellite Images of Xixing Ancient Town from 1969 and 2021 (Drawn by the authors).

## 2. Literature review

After the survey of street space research, this study observed the street life in the ancient town of Xixing through a comparison of these satellite images. Through a quantitative analysis using the spatial syntax method, the study investigated different characteristics of the ancient town, including the structure of streets at different spatial scales and the studies on the renewal strategy of ancient towns with canals can be divided into three aspects: the study of street spatial form, the study of street spatial behavior, and the analysis of street spatial structure.

### 2.1 Research on street spatial form

Several researchers have examined the features and characteristics of street forms from the standpoint of morphology and reached relevant conclusions. Among them, Kevin Lynch (1964) discussed from a cognitive standpoint how the structure, corners, and intersection forms of roads could form different degrees of perception [5]. Ashigara Yoshinobu (1983) focused on the figure-ground relation of street composition, emphasized people's feelings, and advocated the recognizable space rooted in the place [6]. Alan B. Jacobs (1993) recorded the spatial form of many streets in the world, summarized typical street patterns, and compared and summarized the features that great streets should have [7]. All these studies have far-reaching implications. In recent years, many studies have used streetscape image and image segmentation technology, ArcGIS, Internet LBS data, and other means to obtain more specific, more intuitive, and more continuous spatial morphology data of streets to promote the quantitative study of spatial forms of streets and alleys. For example, Doersch et al. (2012) and others employed computer vision and in-depth learning to study the classification of visual elements and architectural styles in Paris by using street scenes [8]. Li et al. (2015) and colleagues used Google Street View images to analyze and assess the greening levels of 300 sample points along a Manhattan block [9]. Tang and Long (2019) explored a new method for visual quality assessment and variation recognition of large area street space, which assesses the physical and perceived visual quality of street space [10].

### 2.2 Street spatial behavior research

Jane Jacobs (1993) observed and studied the interaction of people and street space, and summarized the conditions required for a viable street [11]. Jan Gehl (1971) emphasized the importance of different types of behavioral activities to the vitality of street space. He discussed the physical conditions of street space itself, its design in terms of security and domain perception, and the impact of sensory stimulation on the activation of behavioral activities [12]. Cliff Moughtin (1971) focused on the form and function of streets and squares, proposed design methods in detail, and encouraged pedestrians to take full use of the streets in various ways, thereby stimulating street vitality [13]. Afterward, subsequent studies presented more meta-perspectives. Burton and Mitchell (2006) investigated street space design from the perspective of vulnerable groups living on the streets [14]. Dičiūnaitė-Rauktienė et al. (2018) employed the COPRAS (Complex Proportional Assessment) method to analyze the attitudes and behavior of the residents toward the pedestrian area in three Lithuanian cities, based on which they proposed the strategies for commercialization of the pedestrian area [15]. Xu and Hu (2020) proposed the concept of healing streets from the perspective of people's perception of the healing effect of streets and completed the construction and demonstration of the healing model of streets [16]. Zhe Wang et al. (2022) observed and analyzed social behaviors and data models of the in stelderly reet space to improve the existing strategies and practices of designing street space [17].

## 2.3 Analysis and research on street spatial structure

Space syntax has become a famous research method for street spatial structure analysis. The analysis of street space by employing the space syntax method can obtain the location and scale of street space and determine the layout and activity line of street space, which provides strong proof for the correlation between the form and vitality of street space. Recently, relevant research has made some breakthroughs in research objects, ideas, and paths. For instance, Michael Oloyede Alabi (2021) discussed the syntactic attributes related to urban core blocks for walking behavior, explained the factors affecting the change in walking ability, and provided strategic suggestions for the design of public transport and pedestrian system [18]. Griffiths and Vaughan (2020) clarified the relationship between space syntax and Historical Geographic Information System (HGIS), analyzed the data of typical cities in the 19th century, and promoted the new development of urban history [19]. Leccese et al. (2020) used space syntax to estimate traffic volume in order to design appropriate lighting infrastructure, with the goal of saving energy and avoiding overly large lighting systems [20]. In China, the study of traditional urban street space based on space syntax is gradually increasing. Duan and Bill (2016) analyzed the historic blocks of Sanfangqixiang (Three Lanes and Seven Alleys) in Fuzhou city by using space syntax and quantitatively analyzed their spatial characteristics from macro, medium, and micro scales to explore the path of recovery of Sanfangqixiang [21]. Based on space syntax theory, Wu Xi (2020) discovered a correlation between the number density of street segments and the local integration of traditional blocks in Guiyang city under different parameter radii and explained the influence mechanism of the spatial form of traditional blocks on the vitality of blocks [22]. With the help of spatial syntax, Wang and Guan (2019) took the Guyao District of Jingdezhen City as the research object to explore the correlation and differences between tourists and residents in utilizing street space and conducting activities [23].

To sum up, the research on street space has experienced a long period of development. There is currently a trend toward research on specific topics, cross-domain perspectives, and the use of both qualitative and quantitative methods. The use of spatial syntax analysis of urban street space is one of them, and it has a significant impact on this paper. Still, most of the existing studies are oriented to the internal traffic system of a single town, and few consider the research object as a part of the city itself, which lacks objectivity.

# 3. Research design

## 3.1 Study area

As shown, the core protection area of Xixing ancient town is located within the red-line frame, amounting to 10.49 hectares, and the planning control area is within the blue-line frame, amounting to 25 hectares. In this area, the total area covered by buildings is 223,000 $m^2$, with a building density of 34% and a plot ratio of about 0.9%. Whereas in the core protection area, the total area covered by buildings is 91,000 $m^2$, the building density is 36%, and the plot ratio is 0.7%. In ancient towns, residential land was the major land-use type, which was roughly divided into two categories. The first is low-rise residential buildings, usually traditional, accounting for 66% of the residential land. The second is the recent multi-story residential buildings, accounting for 34% of the residential land. The representative residential areas are Tieling garden located at Tielingguantou, Xingsheng garden at the intersection of Gutang Road and Guling Road, and Guanhexincun near the experimental elementary school at the northeast corner of the block. The scope of this study extends to the west part of Tieling garden from the west, Guanhe apartment in the East, Qingnian road in the south, Gutang road in the

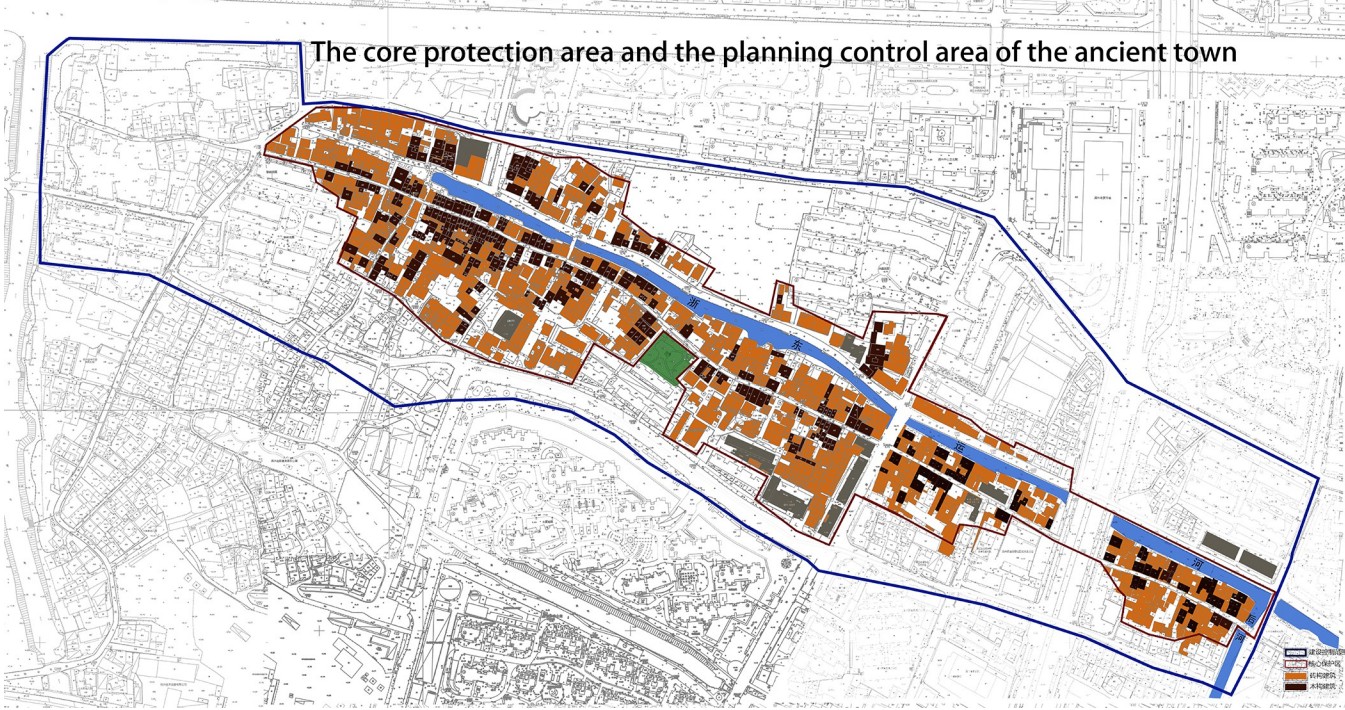

**Fig 3. The core protection area and the planning control area of the ancient town (Drawn by the authors).**

north, and an additional area extending 3 kilometers outwards from the border of the before-said area (see Fig 3).

After correcting the Cad electronic map of Xixing town through field investigation and comparison, the street network is abstracted as the linear model of the minimum and the longest straight lines. The spatial integration under the two scales of core protected area and planning control area is analyzed and compared using Depthmap software, and whether the Zhedong canal has control and influence over the space of Xixing Town is investigated. Buffer zones are defined as an area 3 kilometers out from the planning control scope's border that is within a 40-minute walk distance. Considering that the major ways of transportation between the ancient town and the outside are pedestrian and public transportation, and the boundary effect of the ancient town has been eliminated within 3 kilometers, the ancient town and the city are relatively well integrated. Thus, the integration between the ancient town and the urban space can be better investigated on a larger scale. Based on the topological structure characteristics of streets and alleys in Xixing ancient town, the calculation results such as global integration, R3 local integration, choice, and comprehensibility are obtained to study the internal relationship between streets in the ancient town (especially adjacent streets in the study area) and urban space.

## 3.2 Methodology

In the late 1970s, Bill Hillier and Julienne Hanson of Barrett College, University of London, developed the space syntax method. It is used to explain the spatial forms of various dimensions and socio-economic activities of buildings, communities, and towns from the perspective of space construction activities [24]. After years of practice and corrections made by many case studies, the space syntax theory has developed into a new empirical analysis. It can be

empirically implemented by establishing the standard space syntax axis, line segments, or the viewshed analysis model before being professionally calculated using the DepthMap software to explore the social operation logic behind the spatial relationship via the inter-spatial topological configuration.

Three different spatial analysis methods are derived from spatial syntax, according to different spatial segmentation methods. When the space is linearly distributed, the axis method is used. When the space is distributed in node shape, the convex space method is used, and when there is an obvious occlusion in the space, the horizon segmentation method is used [25]. Based on the former field investigation and the obtained CAD topographic map, this study establishes an axis model of the space syntax according to the standard of "longest and least" before it is processed and analyzed by using the DepthMap software, which interprets the spatial data after the spatial syntax data, such as the integration,choice, and comprehensibility are obtained.

## 4. Result analysis

### 4.1 Spatial integration of ancient towns

The integration is considered as the most important data parameter in the spatial syntax method because it represents the potential of a certain spatial unit to attract the traffic flow, which can be used as a standard to measure the reachability within the whole space. Integration core refers to the first 10%, 25% or 50% of all data, which is considered the most integrated spatial composition mode and an expression of spatial network density. Because the spatial extent of Xixing ancient town is not large and complex as the city network, 10% of the data was used for its study. Moreover, it predicts crowd aggregation when a certain spatial unit of the whole space system is described as a goal space. In terms of the data presentation of integration, the more powerful and warmer the color is, the higher integration it represents. On the contrary, the lower integration it represents, the less powerful and colder the color is [26].

First, the core reserve area of the Xixing ancient town is analyzed using the axis analysis method with the DepthMap software. The canal is calculated as part of the road network and excluded from the road network before the results are obtained and compared. Consequently, the area with the highest integration in the core reserve area of the ancient town is obtained to examine the canal's potential impact on the transportation accessibility of the core reserve area of the ancient town. As shown, if the canal is excluded from the road network and calculated accordingly, the region with the highest global integration is represented by red, and the degree decreases as the color sequence is indicated by red, yellow, green, and blue. If the Zhedong canal is marked as the axis, on the north side of the canal, the integration reaches the peak from the "Tielingguan of Guanhe road to the middle and rear section of the Guling road," whereas on the south side of the canal, the integration reaches the peak around the middle and rear section of the Xixing street and the Guling road that is in the vertical relationship with the Xixing street. Considering the Guling road as the boundary, the degree ranges between 0.382 and 1.305, and the integration of the west side of the ancient town is generally estimated higher than that of the east side. On the contrary, the lowest integration in the ancient town is observed on the southern part of Xixing street where a large number of residential buildings exist with dense lanes and poor accessibility.

When the canal is included and calculated as the road network, since the river ports and public docks on both sides also participate in the calculation, which leads to a higher connection between the two sides of the canal, the highest integration in the ancient town is observed from the Tielingguan of the Guanhe road to the middle and rear section of the Guling road on the northern side of the canal. On the southern side of the canal, the Guanhe road and the

canal area are marked with the highest integration. Considering the Guling road as the boundary, the integration on the east side of the Guanhe road is also higher than that when the canal is not included in the calculation, and the degree ranges from 0.387 to 1.451(see Fig 4).The comparative analysis can be concluded as follows. (1) Regardless of the participation of the Zhedong canal in the calculation as part of the road network, the area with the highest integration is always located on both sides of the canal, which indicates the constancy and stability of the core area of the ancient town. (2) The canal played an important role in the accessibility and connectivity of the core space of the ancient town in the past when the waterway was used as the major traffic approach. (3) As the transportation mode changes, the core of the integration is transferred from the middle section of the canal to the Guling road, which connects the external roads of the ancient town, and the core area of the ancient town is more closely connected with the urban roads.

Within the scope of planning control, when the canal is included in the calculation (Fig 3), the area with high integration moves from the Guanhe road and the Xixing street (from Tielingguan to the middle section of the Guling road) to the Gulin road, Qingnian road, and some segments of the Gutang road. In fact, the highest integration area of the canal still remains from Tielingguan to the middle section of the Guling road, and the degree ranges from 0.433 to 1.710. If the canal is not included in the calculation, the value decreases from 0.432 to 1.669 (see Fig 5). By comparing the spatial integration of the core area and the planning control area of the ancient town, it is observed that (1) as the calculation area expands, the spatial integration is overall improved in the ancient town—the living streets and riverside markets are gradually transferred from the internal roads to the external trunk roads connecting the city, which increases the accessibility and the connection with urban traffic; (2) the canal, being the axis and the road network in either the core protection area or the planning control area, exerts a great impact on the integration of the ancient town.

Through the comparative analysis of the charts and graphs, we can draw the following conclusions: Firstly, at two different spatial scales, independent of the presence of the Zhedong canal in the road network, the areas with the highest degree of integration are always located on both sides of the canal, indicating the stability of the core area of the ancient town. Secondly, after the canal was added to the core protection area, the integration degree was much higher than that without the canal, which indicates that on a small scale, the canal had a higher degree of control over the spatial structure of the ancient town. Thirdly, when the spatial scale is extended to the planning control area, independent of, including the canal for calculation, the value of integration degree does not change significantly, but the overall degree of spatial integration of the ancient town is improved. Thus, it can be concluded that on a large scale, the living streets and street markets along the canal in the ancient town gradually shifted from internal roads to external major roads connecting the new urban areas, increasing the accessibility and connection with urban traffic (see Table 1).

## 4.2 Configuration features of the Xixing ancient town and urban road network

As shown in Fig 4, a buffer zone of 3 km within the planning control range is included in the calculation of the global integration in order to better investigate the spatial topological relationship between ancient town streets and urban road networks. The global integration refers to the accessibility of a single space in the whole space system when the overall system space is considered [27]. On the graph of the global integration, where "r" is equal to "n," the average global integration of the Xixing ancient town is 1.042, and the maximum global integration is 1.763 (see Fig 6). It can be concluded that the core of the integration of the Xixing ancient

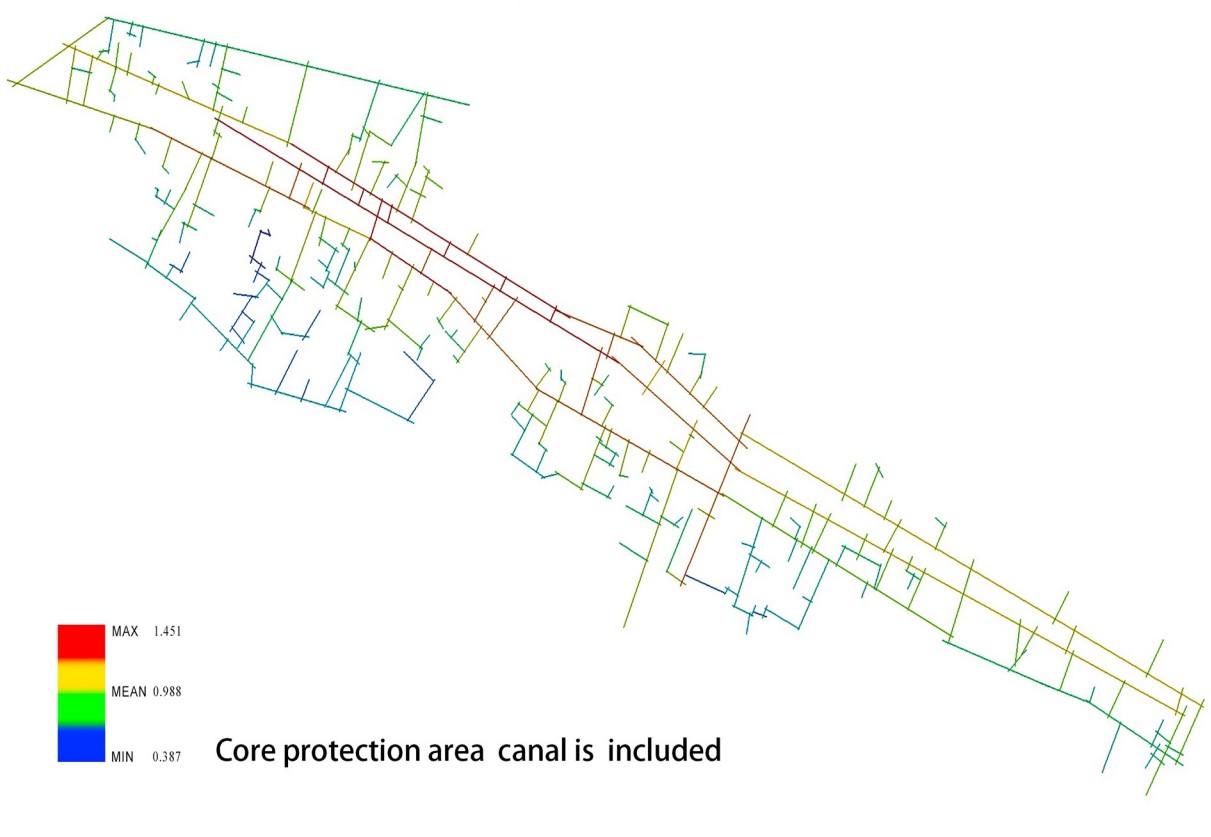

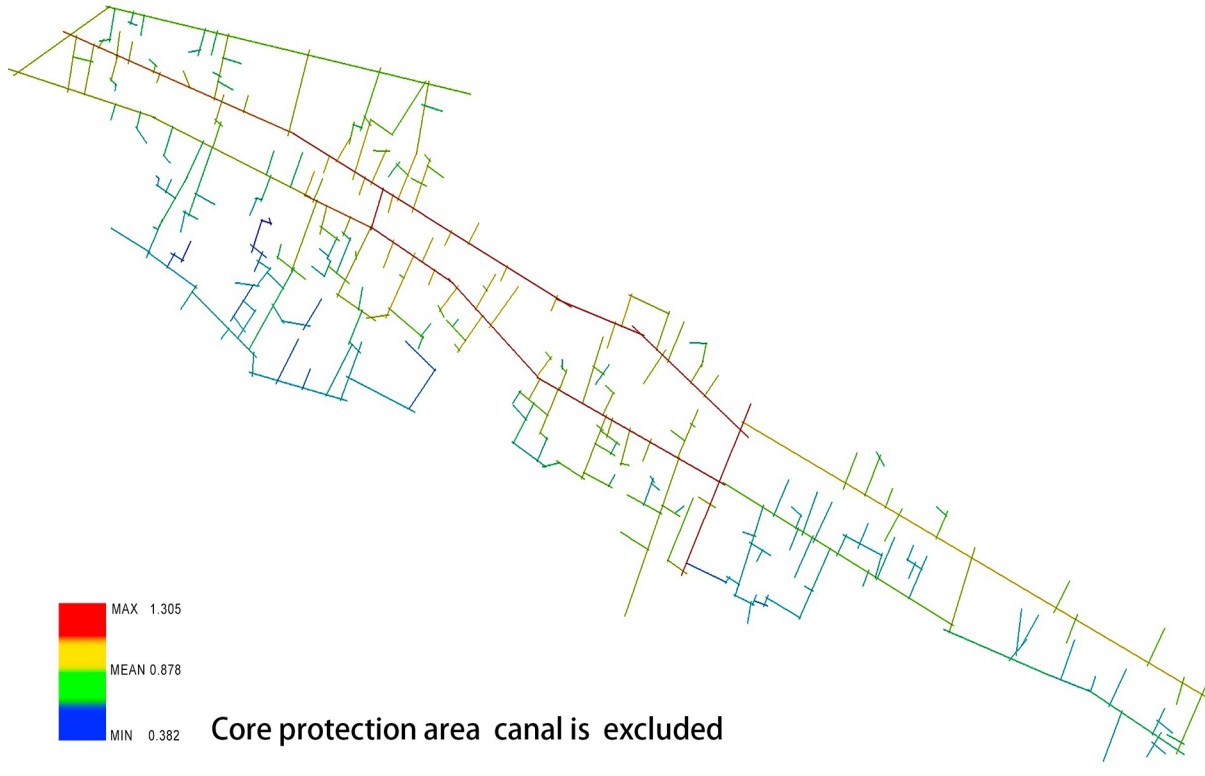

**Fig 4.** (a) Canal is excluded; (b) Canal is included (Drawn by the authors).

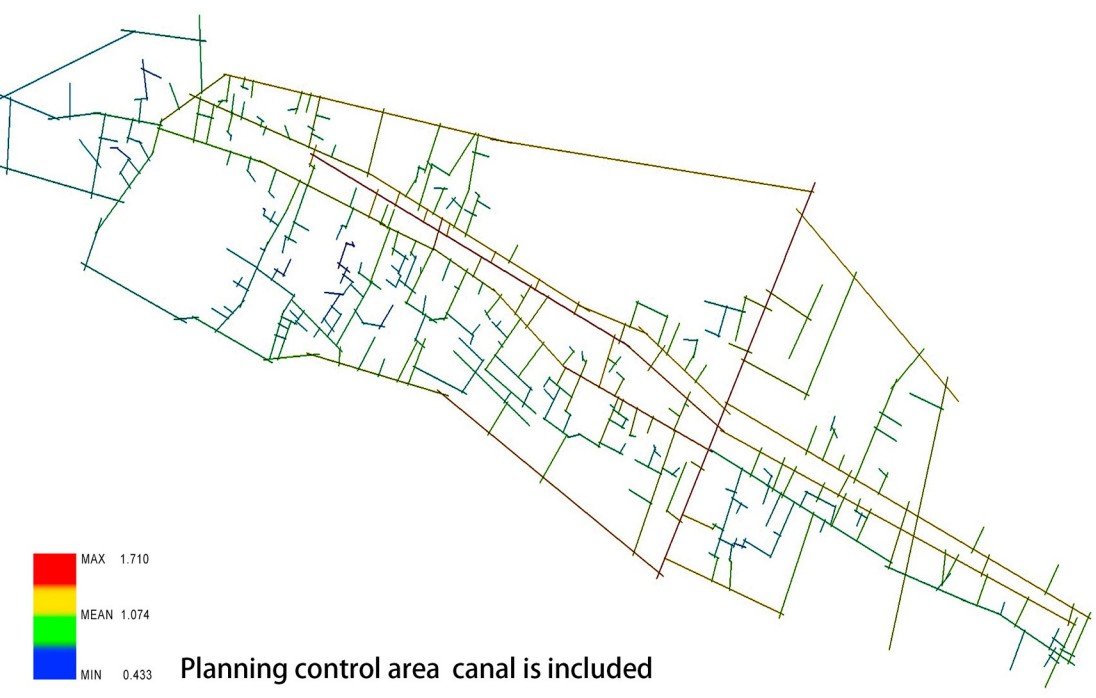

MAX 1.710
MEAN 1.074
MIN 0.433

Planning control area canal is included

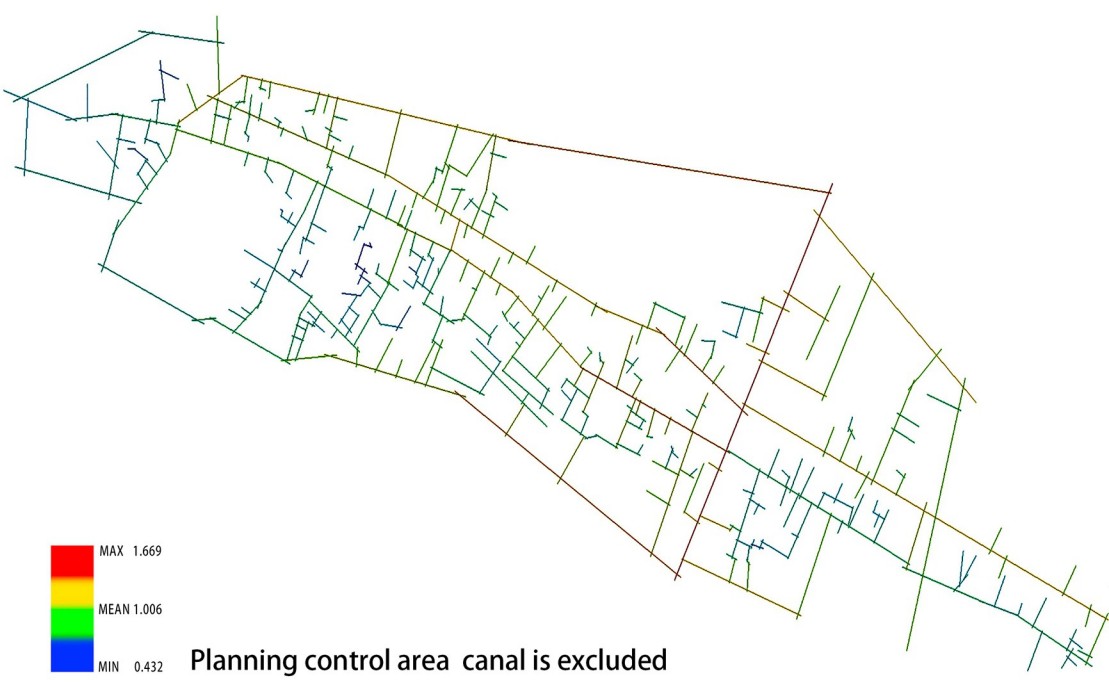

MAX 1.669
MEAN 1.006
MIN 0.432

Planning control area canal is excluded

**Fig 5.** (a) Canal is excluded; (b) Canal is included (Drawn by the authors).

**Table 1. Analysis on integration of ancient towns (Drawn by the authors).**

| Region/Keyword | Canal is excluded/Parameter | Canal is included/Parameter |
|---|---|---|
| Core Protection Area | Mean 0.878 | Mean 0.988 |
| | Max 1.305 | Max 1.451 |
| | Min 0.382 | Min 0.387 |
| Planning Control Area | Mean 1.006 | Mean 1.074 |
| | Max 1.669 | Max 1.710 |
| | Min 0.432 | Min 0.433 |

town includes the areas, including the Beitang road, the Bin'an road, the Xixing road, and the Guling road (the top 5% roads with the integration from high to low). The above-mentioned roads have the highest accessibility and better ability to attract the traffic flow. From the perspective of the functionality, the Bin'an road, as the main road of the Xixing ancient town, is the intersection of the ground traffic and the underground traffic by connecting Hangzhou Metro Line 1. However, the main business district and the agricultural and sideline product market of the Xixing ancient town are the locations that are enclosed by the Guling road, the Beitang road, and the Xixing road. These areas, although with lower accessibility and integration, are easy to form a quiet environment, such as residential quarters and schools.

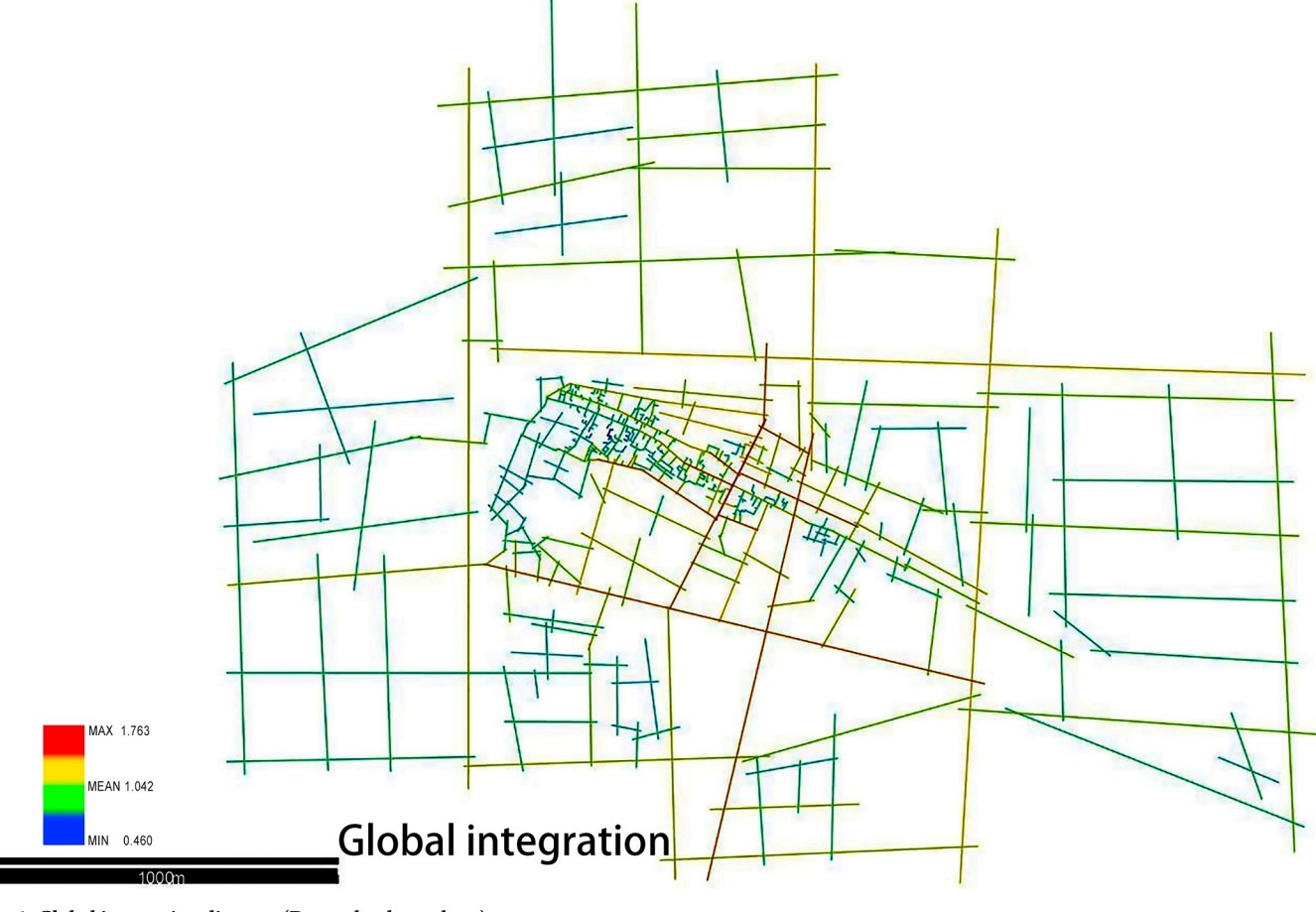

**Fig 6. Global integration diagram (Drawn by the authors).**

According to the space syntax theory, the maximum and average values of the global integration, respectively, represent the accessibility of the foreground network and the background network in the whole space. The larger the value is, the better the accessibility is. The foreground network of the Xixing ancient town (the top 20% roads with the highest integration value) is located in the main traffic trunk roads of the city, such as the Jiangling road, the Xixing road, and the Beitang road. The Guling road that possesses the highest integration is considered the most accessible area of the Xixing ancient town [28].

### 4.3 Local integration degree of the Xixing ancient town

The local integration represents the space steps required to reach the nearby space from a certain space. The R3 local integration refers to the total number of spatial scales that can be achieved with three spatial steps. The higher local integration means more space scale that can be obtained, which reflects the aggregation of space in the localized scale [29].

By using the DepthMap software, the local integration of R3 is obtained, and the average value is about 1.527, and the maximum value is 3.218. As shown in Fig 5, the streets with the highest R3 integration are observed around the major streets of the Xixing town, such as Jiangling road, Bin'an road, Xixing road, and Binkang road, and these streets connect the major functional areas and the main urban trunk roads of the city. On the contrary, the streets with lower R3 integration are the areas such as residential and educational blocks that are tranquil but poor in gathering the crowd (see Fig 7).

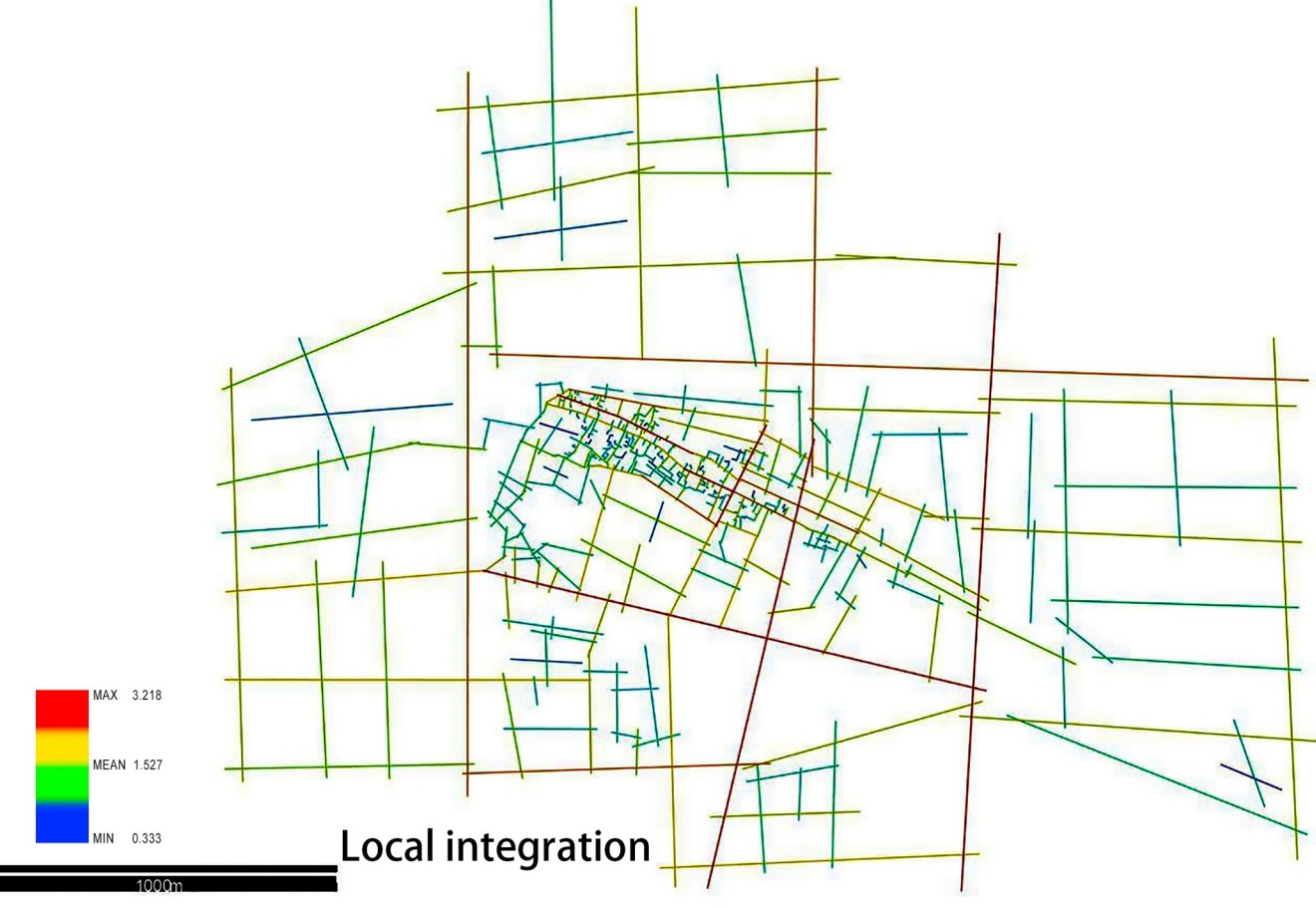

**Fig 7. Local integration (Drawn by the authors).**

**Table 2. Analysis on the integration of ancient town and city (Drawn by the authors).**

| Area | Keyword | Data |
|---|---|---|
| The Whole Area Expanding 3 km Outward from Planning Control Area | Global Integration Degree | Mean 1.042 |
| | | Max 1.763 |
| | | Min 0.460 |
| | Local Integration Degree | Mean 1.527 |
| | | Max 3.218 |
| | | Min 0.333 |

By analyzing the characteristics of the R3 local integration, the integration is centrally located in the trunk roads such as Jiangling road, Bin'an road, Xixing road, and Fengqing avenue, which connect the external traffic and the main internal roads of the ancient town.

The general integration degree of the whole area expanding 3 kilometers outward from planning control area and the local integration degree of the planning control area were compared to analyze the road network in the ancient town. It was concluded that on a larger scale, the ancient town is more closely connected with the urban road network system than the inner adjacent streets and lanes. The main roads in the cities are important as they lay the foundation for the integration of ancient towns into cities and for better sustainable development (see Table 2).

## 4.4 The choice of the Xixing ancient town

In space syntax theory, the choice represents the potential of space to be traversed in the whole unit. The greater choice means, the greater potential to be traversed [30]. The value of the choice shown in Fig 6 indicates an obvious disparity between the foreground and the background road network of the town. The street with the highest choice appears in the Guling road, and the other high values are observed in the Xixing road and the Bin'an road that are the major roads connecting the urban space of the ancient town and they belong to the foreground road network of the town. Inside the ancient town, the Xixing street and the Guanhe road show a higher choice value, which means they have greater potential to be traversed as a first choice (see Fig 8, see Table 3).

Actually, Xixing street and the Guanhe road are quite friendly and mostly used by pedestrians since their width is about 1.5–3.5 m. However, several urban foreground roads are designed as two-way roads with four lanes, which are used for the transportation between cities and towns. Considering the Guling road as an example, the safety of vehicles and people is ensured by adding guardrails and dividers between them. As compared to the calculated results of the local choice, the value of the global choice resembles that of the local choice, which indicates an unobvious difference in the traffic distribution; thus, making it useless to design alternatively for pedestrians and vehicles, so the original traffic planning and the management system are still operational.

## 4.5 The intelligibility of the Xixing ancient town

In the intelligibility graph, the abscissa x represents the connection value, the ordinate y represents the global integration, the oblique line represents the trend line, and $R^2$ represents the fitting. According to mathematical statistics, the higher the $R^2$ value is, the more accurate the regression line is in predicting the actual situation of the scatter diagram. Generally speaking, when $R^2$ is below 0.5, it shows a relatively low intelligibility. When $R^2$ is above 0.5, the intelligibility of the relative space is relatively high. As indicated in Fig 7, the value of intelligibility $R^2$ is 0.255, which is far less than the measurement value of 0.5. When the value in the figure is

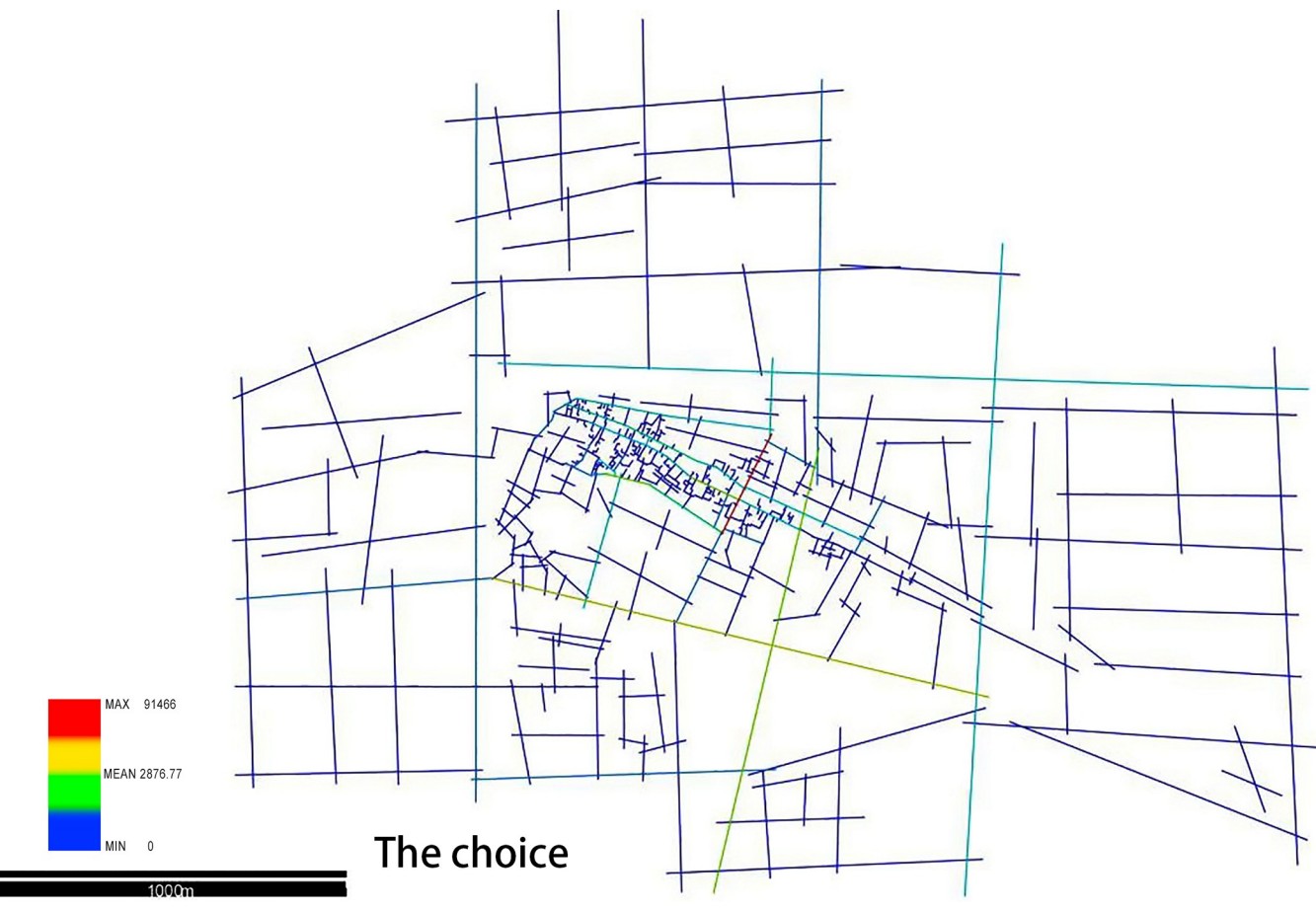

**Fig 8. The choice (Drawn by the authors).**

accurately examined, it can be noted that most of the street segments are located in the end space of the street or the remote end of the road space, which are rarely located in the street space with much convenience, thus leading to an unfavorable accommodation between the partial space and the whole space, which makes the space lacking in clarity (see Fig 9).

Presently, the streets and alleys in the Xixing historical district have not been spatially updated and arranged. With regard to intelligibility, different levels of streets, lanes, and alleys in the ancient town are observed with low connectivity, and the streets and alleys are often seen with dead ends. Consequently, it is noted, based on this phenomenon, that modern asphalt streets (core areas of the block) show a higher integration, and a lower integration is observed in the internal roads of historical blocks (blind areas of the block). This kind of spatial structure not only affects the tourists' perception and recognition of space but also hinders the convenient transportation of local residents and breaks up the connection with the urban space.

**Table 3. Analysis on the choice of ancient town and city (Drawn by the authors).**

| Area | Keyword | Data |
|---|---|---|
| The Whole Area Expanding 3 km Outward from Planning Control Area | Choice | Mean 2876.77 |
| | | Max 91466 |
| | | Min 0 |

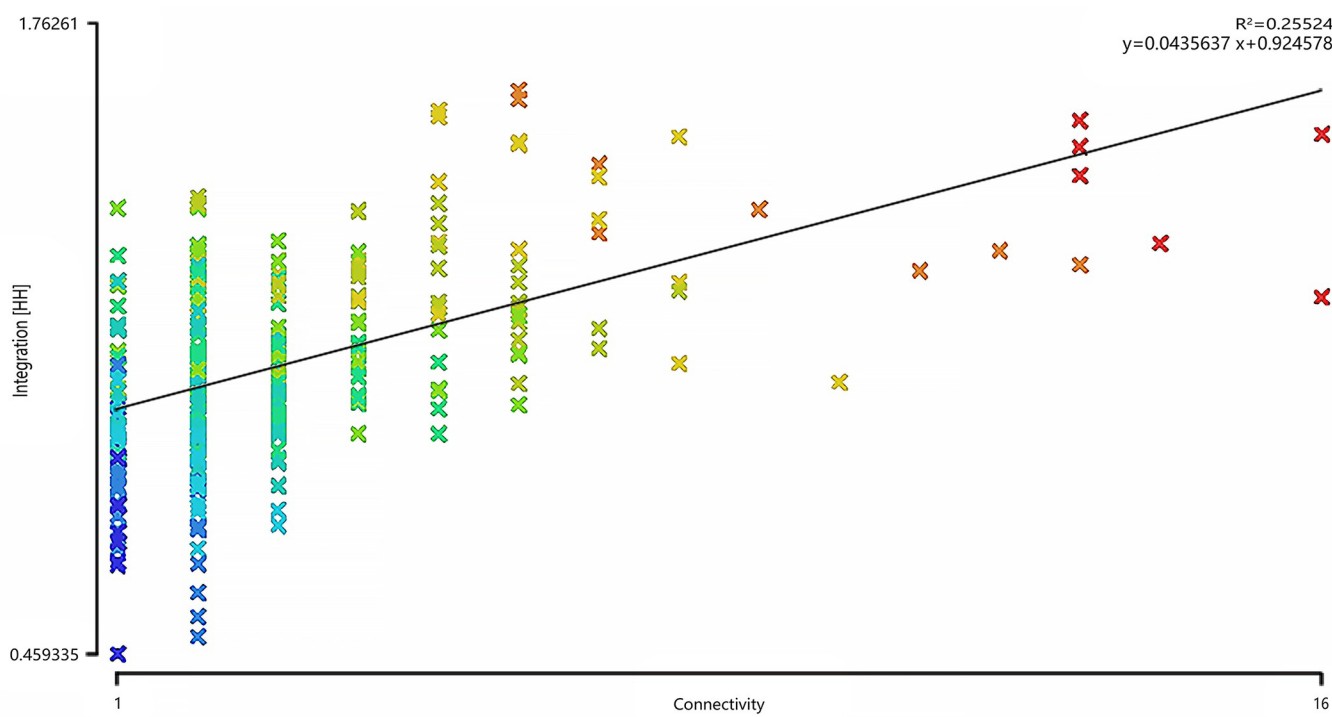

Fig 9. Scatter diagram of the intelligibility (Drawn by the authors).

## 5. Results

Based on the theory and method of spatial syntax, this paper studied the spatial and topological configuration of streets and lanes in Xixing ancient town and its spatial organization relationship with the urban system. The study reveals that when the ancient town is placed on a larger urban scale, the area with the best integration and accessibility of the ancient town changes on both sides of the canal to the intersection space and urban road network. Specifically, to realize the living and sustainable development of the ancient town, and additionally to renew the space and function of the ancient town itself, it is necessary to incorporate, the ancient town into the urban system and maintain its continuity and connectivity with the urban space. Moreover, the evaluation and prediction of spatial syntax can help the ancient town better realize the improvement of average land sharing, commercial distribution, public facilities layout and other aspects in the practice of protection and development.

## 6. Discussion

A comparison of satellite maps of ancient towns from the late 1960s with those from the 1920s show that the water system and road network around the canal increased in the recent years. The areas with high integration and accessibility of ancient towns gradually extended from the inner streets of towns to the urban streets, but the ancient town's core area remained on both sides of the canal. This type of isolated self-development of ancient towns with canals is easy to cause the splitting of the towns and urban space. This does not respond well to the concept that the residential settlement of ancient towns with canals should be taken as integral protection of ancient town groups that was proposed in the Outline of Planning for the

Conservation, Inheritance, and Utilization of Grand Canal Culture. Specifically, to realize the sustainable development of ancient towns, the endogenous resilience of ancient towns and the synergistic integration with cities should be fully considered.

Based on the parametric calculation of spatial syntax, combined with the existing spatial morphological characteristics of Xixing ancient town, the following three strategies are proposed.

## 6.1 The optimization of links with Urban space systems

Ancient towns with canals belong to medium and small-scale space compared to urban space. To build a 3 kilometers wide transition buffer between Xixing ancient town and the urban space network which would act as a coordination zone, The setting up of this area should focus on optimizing the connection between the ancient town with canals and the urban transportation system, evenly distributing the land usage, completing the functional industries, so as to promote the better integration and development of the ancient town with the urban area, and stimulate the sustainable endogenous forces.

## 6.2 Establishment of a multi-level transportation system composed of vehicle and pedestrian

For the renewal of the space of streets and alleys in the ancient towns with canals, the first step is to establish a scientific small-scale spatial transportation system for the residential blocks. At the intersection of ancient towns and urban roads, on the basis of conforming to the living style of modern people, the density distribution of various levels of entrances and exits, branches, roadways, and other modes of travel should be rationally arranged. The second is the connection with the urban road network system. Three-dimensional transportation network can be constructed on the existing basis, including public parking lot, small tourist centers, and other transportation facilities, including setting in the coordination zone to increase the quick connection between the ancient town and the city. Third, to scientifically solve the mixed traffic mode of motor vehicles and various non-motor vehicles. Appropriate parking areas for motor vehicles and non-motor vehicles can be set up within the planning control area of the ancient town to facilitate the residents of the ancient town and to better achieve the traffic synergy at different levels of the ancient town. Furthermore, in order to restore the liveliness of the landscape and public space in Xixing ancient town, it is necessary to consolidate the fragmented space, construct the landscape and safety areas required for the implementation of these activities, and improve the quality of life and attraction for investment in the area [31].

## 6.3 Construction of the adaptive mode of the integration of living environment and business space

In different historical periods, ancient towns with canals evolved in accordance with the requirements of economic and social development. For the ancient towns with canals where business model is updated, the characteristics of commercial spatial distribution along streets and rivers are effectively analyzed. Regression analysis of spatial data such as the number of commercial entities on each street segment, the area, area width, depth of single building and spatial syntax parameters is carried out. The analysis can be used to quantify and predict the commercial distribution after the update of the street network. For Xixing ancient town, on the one hand, it can using Baidu POI data, the distribution of two commercial functions on the scale of residential blocks is attempted. One is to analyze the commercial distribution data

of street sections through the commercial flow analysis method. In view of the total amount of all kinds of commercial entities in spite of their forms within the street segments, the regression analysis method is used to explore the relationship between the amount and various parameters of space syntax. The second is to focus on the spatial distribution rule of a specific commercial form, especially the distribution of commercial convenience. The average distance and road accessibility grade of the distribution of this specific business form are used for statistical analysis. Through this, the distribution of community service functions can be increased. On the other hand, in order to create historical and cultural blocks, digitalization and smart-city approaches can be integrated into the governance of space to establish a sustainable development model of tourism town space [32], which is also an effective means to realize the space revival of Xixing ancient town.

## 7. Conclusion and expectations

Based on the theory and method of spatial syntax, this paper studied the spatial and topological configuration of streets and lanes in Xixing ancient town and its spatial organization relationship with the urban system. The study reveals that when the ancient town is placed on a larger urban scale, the area with the best integration and accessibility of the ancient town changes on both sides of the canal to the intersection space and urban road network. This conclusion can be used to guide the practices of protection and renewal of the ancient towns with canals in China. Specifically, to realize the sustainable development of the ancient town, and additionally renew the space and function of the ancient town itself, it is necessary to incorporate it into the urban system and maintain its continuity and connectivity with the urban space. Moreover, the evaluation and prediction of spatial syntax can help the ancient town better realize the improvement of average land sharing, commercial distribution, public facilities layout and other aspects in the practice of protection and development.

So far, this research can continue to make progress in terms of the breadth of perspectives and the depth of analysis. In future research, the ancient canal towns with different scales, spatial forms, and protection and development modes can be evenly compared. In addition, the development and integration of multiple methods will also help us to understand the ancient canal town space in depth. The focus of the study can be placed on the ancient towns along the Zhedong canal, thereby realizing the protection and renewal of the ancient towns through the strategy of "harmony in difference".

## Supporting information

**S1 File.**
(DOCX)

**S1 Data.**
(CSV)

**S2 Data.**
(CSV)

**S3 Data.**
(CSV)

## Acknowledgments

This research model was completed with the assistance and cooperation of Wang Bing. I would like to express my thanks.

## Author Contributions

**Data curation:** Weilu Lv.

**Funding acquisition:** Yun Huang.

**Investigation:** Weilu Lv.

**Methodology:** Weilu Lv.

**Software:** Ning Wang.

**Supervision:** Weilu Lv, Ning Wang, Yun Huang.

**Visualization:** Weilu Lv, Ning Wang, Yun Huang.

**Writing – original draft:** Weilu Lv.

**Writing – review & editing:** Weilu Lv, Ning Wang, Yun Huang.

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
