## [Decision Letter · Decision Letter 0]

18 Jul 2022

PONE-D-22-18649Study on the Sustainable Renewal Strategy of Ancient Towns with Canals on the Basis of the Topological Structure Analysis of Streets and Lanes: A Case Study of the Xixing Ancient TownPLOS ONE

Dear Dr. Wang,

Thank you for submitting your manuscript to PLOS ONE. After careful consideration, we feel that it has merit but does not fully meet PLOS ONE’s publication criteria as it currently stands. Therefore, we invite you to submit a revised version of the manuscript that addresses the points raised during the review process.

We look forward to receiving your revised manuscript.

Kind regards,

Dragan Pamucar

Academic Editor

PLOS ONE

Journal Requirements:

6. We note that Figures 1 - 7 in your submission contain [map/satellite] images which may be copyrighted. All PLOS content is published under the Creative Commons Attribution License (CC BY 4.0), which means that the manuscript, images, and Supporting Information files will be freely available online, and any third party is permitted to access, download, copy, distribute, and use these materials in any way, even commercially, with proper attribution. For these reasons, we cannot publish previously copyrighted maps or satellite images created using proprietary data, such as Google software (Google Maps, Street View, and Earth). For more information, see our copyright guidelines: http://journals.plos.org/plosone/s/licenses-and-copyright.

a. You may seek permission from the original copyright holder of Figures 1 - 7 to publish the content specifically under the CC BY 4.0 license.  

Reviewers' comments:

Reviewer's Responses to Questions

**Comments to the Author**

1. Is the manuscript technically sound, and do the data support the conclusions?

Reviewer #1: Partly

Reviewer #2: No

Reviewer #3: Yes

2. Has the statistical analysis been performed appropriately and rigorously? 

Reviewer #1: Yes

Reviewer #2: No

Reviewer #3: Yes

3. Have the authors made all data underlying the findings in their manuscript fully available?

Reviewer #1: Yes

Reviewer #2: No

Reviewer #3: Yes

4. Is the manuscript presented in an intelligible fashion and written in standard English?

Reviewer #1: Yes

Reviewer #2: No

Reviewer #3: Yes

5. Review Comments to the Author

Reviewer #1: This paper discusses the spatial syntax to analyze the topological structure of roads and alleys in the ancient city of Xixing in the Zhedong canal as a case object. I think this topic is interesting regard to most situations in urban development. But this paper should be improved in order to publishing requirement in the international journal. Specific comments are noted below:

1. Authors(s) already provided the abstract’s content (the purpose/background, method used, results). However, please provide clear methods on the studies that have been conducted. Please make it short and sharp. In addition, I recommend writing a clear statement about methodology. It is not necessary to write the steps of the method, but briefly explain the method used, analysis, and tools used. Write the keywords in alphabetical order.

2. In the initial paragraph in the introduction the problem is less clear from the author's point of view. It should refer after summarizing some of the results from the existing references. Some sentences that need to be corrected such as "...As a result, how to conduct innovative research on the space protection and vitality renewal of ancient towns with canals in the new historical period has become an important topic for the continuity and long-term development of those ancient towns in the modern era…” Further explanation is needed in this statement. The introduction is something that explains the contents of the entire manuscript. I suggest entering the contribution section in the introduction (contribution is part of introduction). Make the sentences flow and adjust the sentence by referring to the reference described in the initial paragraph in the introduction.

3. Did you mean Yoshinobu Ashihara? (line 77).

4. Literature review is a section that explains the basis of the study conducted. It is necessary to explain which parts of the reference are used in the methodology and then used to carry out the analysis. How the author(s) explains this into the methodology is important. It is highly recommended to summarize each part of the literature review by relating it to the research topic being carried out. State at the end of each paragraph each literature review (lines 90, 104, and 127).

5. Paper for international journals will be read by readers who do not understand the location, so give information in the form of maps of Xixing, China, and which parts of China are studied.

6. Explain with a statement at the end of the discussion section that explains the analysis of the discussion that has been described. Therefore, it is necessary to distinguish which part of the results and which part of the discussion. Make it in its own part.

7. Conclusion is the answer to the question asked in the introduction. So, the conclusion in this article should answer what is being questioned. Need to do a sharper explanation is actually what problems to be answered. In my opinion, there is no conclusion and it is too long for conclusion section, but it is also necessary to answer that should be at the end of the introduction. Please avoid writing conclusions in points, make in paragraph sentences. Two to three paragraphs for the conclusion section are sufficient.

8. Each figure requires a caption and explanation in the image, for example adding text to the captured image calculation from DepthMap.

9. In general, systematics in writing of scientific articles is Introduction, Literature review, Research Methods, Results and Discussion, and Conclusions. Systematics should be used in writing articles in this paper need to be improved.

This paper actually is an interesting topic to be raised; the author(s) tries to contribute the study regarding spatial syntax in the form of integration, choice, synergy, and clarity of streets and alleys, in order to obtain an update strategy and conclusions about the spatial vitality of a city. But this paper requires adjustment to the structure of a journal; it is necessary to follow the structure of the existing journal writing in general, so that the flow of writing becomes easy to understand.

Reviewer #2: Although the authors have attempted to address the reviewers' comments provided in the previous round of review, I still think the paper is far below the publishable standard for a top-tier journal. Critical issues to have addressed include:

1) The title of the paper is too wordy. It should have been revised.

2) The choice of the study area has not been well justified.

3) The paper is still not adequately referenced. For example, sources of information is missing for lines 136-148. References are also needed for lines 165-166. Many other places in the paper have missing references.

4) A clear conceptual framework for the research is missing in the paper.

5) The theme of the paper should rest on renewal of ancient towns. However, the literature review does not cover much literature related to that area.

6) Why was the "space syntax theory" chosen? Why not other theories or models?

7) What are the limitations of the research? How do these limitations affect the interpretations of the research findings?

8) The paper has a lot typos and grammatical errors. The authors should have the paper proofread by a professional English writer before submission.

Reviewer #3: The manuscript deals with an interesting research topic, having a well organized but poorly addressed research study. Therefore, there is room for significant methodological, morphological, and argumentation improvements to be made, prior it to be accepted for publication at the PLOS ONE journal. To this end the following review comments can be considered.

1)A rough estimation of chronological flash-back to the century of the Xixing ancient town’ foundation, it can be provided in the Abstract section (in approximate century B.C.).

2)In case of some input data or results derived from the analysis, then, a selected and representative portion of them they can be collectively denoted in Abstract section, enabling a more precise representation of the relevant analysis and findings in the Abstract extract, accordingly.

3)At the Introduction section authors are recommended to reorganize the relevant theoretical coverage into two distinct and separated subsections, relevant to two subsections/subheadings of: a) Geographical overview of ancient towns in China and b) Chronological overview of ancients towns establishment and evolution to the today Chinese context. The spatial characteristics approached in section 2. Literature Review are well organized. However, a missing information of both narratives of sections Introduction, Literature Review is that there is absolutely missing numerical information-outcomes that could be collectively represented in the form of a Table at the end of section 2. There is not need authors to recall the narrative statements, but to enrich them with numerical information fitting to the already, or to-be-developed, subsections of main sections 1 and 2. These data of 8-10 citations are adequate.

4)At the Methodology section authors are recommended to briefly represent the land uses of the ancient town examined, in terms of: a) agricultural zone and area occupied (in km^2, or as percentage of the total area of the town), urban zones: pedestrian, streets and lanes (in km^2, or as percentage of the total area of the town), irrigation system and rainfall profile (in mm), domesticated area.

5)In my opinion the heading/section of “Research Design” it is part of the Methodology section, thus, it can be transferred in it, accordingly. Besides, the main frame of headings should be only that of (adding those main headings, where missing): 1. Introduction, 2. Literature Review, 3. Methodology, 4. Results, 5. Discussion, 6. Conclusions and Future Research Orientations. Then, other main headings can be degraded to subheadings’ relocated into the aforementioned headings-frame. Moreover, starting from the 1. Introduction and end to 6. Conclusions and Future Research Orientations section, all embodied subheadings can be titled as: 1.1, 1.2, ……., 2.1, 2.2, …..3.1, 3.2, ….., 4.1, ….4.2, ……., in successive order and numbering, not to be plenary given.

6)At the Discussion section should be reorganized focusing on the today reflections and commercial commitments/services offered, in alignment with the stated research objective: “spatial data such as the number of commercial entities on each street segment, the area, area width, depth of single building and spatial syntax parameters is carried out”. To this end the following key-determinants can be briefly outlined: a) visiting the ancient towns in China, as an inseparable part of the today tourism industry, b) commercial, manufacturing and industrial activities and products/goods’ offered, c) the role of technology in the main community uses: agricultural sector, irrigation technologies, manufacturing and public services accommodated, d) role of governmental policies and motivations scheduled in supporting the ancient towns heritage, protection, and development of related to infrastructure at the nearby, to ancient towns, areas in China. To this end the “Further Discussion”:

-It can be detached, “upgraded as main section”, and presented as separated “Discussion” section, since it contains purely argumentative statements, though all of them have to be accompanied by cross-citations that are currently missing.

-It can be accompanied by numerical information, data, quantitative results, enabling a more precise and comprehensive “guidance” of research outcomes, proposals, and recommendations to areas and contexts of the same research- and spatial- characteristics to be derived and applied.

7)Since Figures and all subfigures of 2 up to 6 are graphically liking to each other, it is highly recommended authors to either provide a more detailed information/interpretation of the comparing and contrasting them, revealing what are the distinct characteristics from the one subfigure to the other one. To this end authors can update these graphs by adding indicators of presentation in the form of arrows into these maps, it could be highly supportive and recommended.

8)There is a long arrangement of numbers of 6 or more decimal digits and fluctuated precision, like that of lines 205, 217, 229-230, but also in Tables data. Therefore, revising and fixing the precision of all data to only 3 decimal digits it is highly recommended for the scopes of this manuscript.

9)Figure 7 can be presented in a white-coloured background, in better visual contrast with the trend-line dots to be achieved.

10)Literature enrichment with more and relevant published papers it is recommended, thus, checking and citing among the below listed studies, where missing and where matching to the revised manuscript, it can be undertaken.

Scopus

EXPORT DATE:12 Jul 2022

Wang, Z., Zhang, H., Yang, X., Li, G.

36469470000;57373042100;57373940300;57373581900;

Neighborhood streets as places of older adults’ active travel and social interaction – A study in Daokou ancient town

(2022) Journal of Transport and Health, 24, art. no. 101309, . Cited 1 time.

https://www.scopus.com/inward/record.uri?eid=2-s2.0-85121263625&doi=10.1016%2fj.jth.2021.101309&partnerID=40&md5=708e7f95f3c811844c51f5b25830e74f

DOI: 10.1016/j.jth.2021.101309

AFFILIATIONS: Department of Architecture, Henan University, Kaifeng, China;

Department of Architecture, Henan University of Technology, Zhengzhou, China;

Department of Architecture, Zhengzhou University, Zhengzhou, China

ABSTRACT: Introduction: Neighborhood streets are convenient places for older adults to engage in behaviors for active living, such as walking (active travel) and chatting with neighbors (social interaction). Street environments and older adults' active living in ancient towns need investigation. Taking Daokou ancient town in China as an example, this research observed older adults’ active travel and social interaction on two neighborhood streets and investigated the difference in social engagement between older-adult groups on different streets. Methods: On-site non-participant observation was conducted for four weekdays with seven 30-min sections per day. ...................

Manakina, A., Nikolaeva, V.

56134235500;57218826705;

Formation of sustainable development of bicycle and pedestrian zones in a modern city

(2020) IOP Conference Series: Materials Science and Engineering, 890 (1), art. no. 012179, .

https://www.scopus.com/inward/record.uri?eid=2-s2.0-85090348649&doi=10.1088%2f1757-899X%2f890%2f1%2f012179&partnerID=40&md5=c0bf49b51025943e083d8eaf6c93b826

DOI: 10.1088/1757-899X/890/1/012179

AFFILIATIONS: Industrial University, Tyumen, 625000, Russian Federation

ABSTRACT: The article is devoted to the current theme of bicycle and pedestrian space development in the developing city, as well as to attracting residents to the landscaped parts of the city for active recreation. The authors consider various types of activity, the implementation of which requires landscaped and safe areas, as well as enumerate the positive aspects associated with the introduction of active recreation in the daily life of residents, therefore, improve the quality of life and increase investment attractiveness of the region. ...............

Yu, X., Ma, S., Cheng, K., Kyriakopoulos, G.L.

53865793000;57218326388;57218326647;6603382498;

An evaluation system for sustainable urban space development based in green urbanism principles-a case study based on the Qin-Ba mountain area in China

(2020) Sustainability (Switzerland), 12 (14), art. no. 5703, . Cited 24 times.

https://www.scopus.com/inward/record.uri?eid=2-s2.0-85088828018&doi=10.3390%2fsu12145703&partnerID=40&md5=f4c407b3b707d7b77b1350577b4e98e5

DOI: 10.3390/su12145703

AFFILIATIONS: School of Architecture, Chang'an University, Xi'an, 710061, China;

Electric Power Division, Photometry Laboratory, School of Electrical and Computer Engineering, National Technical University of Athens, Athens, 15780, Greece

ABSTRACT: Since the 20th century, the deterioration of the ecological environment around the world has challenged urban space construction. With the development of urbanization, the consumption of resources and energy has increased, the level of biodiversity has decreased, environmental pollution is approaching the critical level, and the contradiction between human habitat activity and ecological environment has become increasingly prominent. The sustainable development of urban space along with its economic and social benefits, taking into account the quality of life and ecological environment, has become a new and important subject that needs to be explored. In this study, the indices of the evaluation system for sustainable urban spatial development in regions with underdeveloped economies but rich in ecological resources are arranged in sequence through the systematic coupling analysis of collaborative evaluation information and a quantitative analysis. The influences of urban space elements on sustainable urban development are disclosed. On the basis of the generated data, an evaluation system for sustainable urban spatial development with a complete set of information is proposed. The proposed system is applicable to urban spatial development evaluation in regions in China with underdeveloped economies but rich in ecological capital. First, the basic concept of system coupling is introduced, and a coupling relationship between urban sustainable development and urban space is proposed. Second, the elements of urban space and the sustainable development in the Qin-Ba mountain area are extracted, and the precedence diagram method is used to construct a sustainable evaluation system for urban space development in the Qin-Ba mountain area. Third, the sustainable evaluation process of urban spatial development is proposed. Finally, the sustainable evaluation system for urban spatial development in the Qin-Ba mountain area is applied to evaluate the urban spatial development in Shangluo, Qin-Ling Mountains, China. The results show that, among the investigated 14 indicators, the proportion of industrial land use mainly influences sustainable urban spatial development. As for the rest of the index factors, per capita green land area and green coverage ratio of built-up areas, per capita urban construction land area, proportion of forestry area, greening rate of built-up areas, total industrial dust emission density, proportion of cultivated area, and average volume fraction of residential areas are the secondary influencing factors of sustainable urban spatial development. The evaluation system in this research is constructed with the three aspects of "green coordination", "green development", and "green sustainability" of sustainable urban spatial development, and it complements the evaluation contents of urban-rural ecological space coordination, land resource protection, and green development community, and so on...............

Nomura, R., Shan, S., Mori, S.

57191923705;57204313367;37112638600;

Analyzing spatial structure of traditional houses in old towns with tourism development and its transformation toward sustainable development of residential environments in hexia old town, in China

(2018) Sustainability (Switzerland), 10 (10), art. no. 3809, . Cited 6 times.

https://www.scopus.com/inward/record.uri?eid=2-s2.0-85055187483&doi=10.3390%2fsu10103809&partnerID=40&md5=69acac765079e477ed50bff363e6ac4c

DOI: 10.3390/su10103809

AFFILIATIONS: Faculty of Engineering, Hokkaido University, Sapporo, 060-8628, Japan;

Shanghai Urban Construction Design and Research Institute (Group) Co., Ltd., Shanghai, 200125, China

ABSTRACT: This study elucidates the spatial structure of traditional houses, and its transformation toward sustainable development of residential environments in old towns that are currently the focus of tourism development initiatives. Hexia old town in Huai'an District, China, was chosen for this study involving a field survey, questionnaire survey, and interviews. Data analysis identified three periods of residential transformation and three patterns of the residential transformation process. In all cases, there was low awareness of building conservation measures and lack of tourism development by the government, which has translated into ongoing residential transformations according to the demands of daily life. However, in the interests of tourism development, some businesses have started making use of vacant residences and traditional homes, and the relative proximity to work and home imply sustainability of residents' lifestyles under these conditions. .............

Dičiunaite-Rauktiene, R., Gurskiene, V., Burinskiene, M., Maliene, V.

57201215996;8963424300;10040854900;24332653600;

The usage and perception of pedestrian zones in Lithuanian Cities: Multiple Criteria and comparative analysis

(2018) Sustainability (Switzerland), 10 (3), art. no. 818, . Cited 9 times.

https://www.scopus.com/inward/record.uri?eid=2-s2.0-85044007619&doi=10.3390%2fsu10030818&partnerID=40&md5=e9d368d976fc11b7515f76720018ed38

DOI: 10.3390/su10030818

AFFILIATIONS: Institute of Land Management and Geomatics, Faculty ofWater and Land Management, Aleksandras Stulginskis University, Studentu 11, Akademija, Kaunas distr., LT-53361, Lithuania;

Road Department, Faculty of Environmental Engineering, Vilnius Gediminas Technical University, Sauletekio ave. 11, Vilnius, LT-10223, Lithuania;

Department of the Built Environment, The Built Environment and Sustainable Technologies Research Institute, Faculty of Engineering and Technology, Liverpool John Moores University, Byrom street, Liverpool, L3 3AF, United Kingdom

ABSTRACT: As pedestrian zones are public spaces in urban areas, they are important in terms of meeting people's needs. However, it is worth noting that attention should be paid not only to the development of the physical infrastructure, but also to a sustainable balance between the socio-economic and environmental aspects. To guide urban planning and management initiatives towards more sustainable patterns, it is essential to re-examine the already existing characteristics of cities, establishing how they are used and perceived by inhabitants. The present research suggests environmental, economic and social criteria that determine a greater vitality of pedestrian zones in cities and better life quality for the inhabitants. A questionnaire survey was used to assess common attitudes regarding the research topic in major cities in Lithuania. The multiple criteria decision-making COPRAS (COmplex PRoportional ASsessment) method was used for the formation of a priority queue. ...............

Yang, F.

36607241700;

Investigating wintertime pedestrian wind environment and user perception in dense residential neighbourhood in a city of hot-summer and cold-winter climate zone, China

(2017) Indoor and Built Environment, 26 (3), pp. 392-408. Cited 3 times.

https://www.scopus.com/inward/record.uri?eid=2-s2.0-85018181386&doi=10.1177%2f1420326X15620257&partnerID=40&md5=3afd66a42594b9f0e83e188c3d22fbdc

DOI: 10.1177/1420326X15620257

AFFILIATIONS: College of Architecture and Urban Planning (CAUP), Tongji University, 1239 Siping Road, Shanghai, 200092, China;

Key Laboratory of Ecology and Energy-Saving Study of Dense Habitat, Tongji University, Ministry of Education, Shanghai, China

ABSTRACT: Urban wind environment could have an impact on pedestrian's comfort and safety, as well as pollution dispersion and building energy consumption. For cities in the hot-summer and cold-winter climate zone of China, a proper design residential neighbourhoods is important to facilitate urban ventilation in hot and transient seasons and to protect users from strong winds in cold season. This paper reports the results of field measurements and a questionnaire survey in a large residential development with three different types of housings. Micrometeorology measurement was carried out at the pedestrian height level as well as at a rooftop reference station. Pedestrians' subjective perception on wind and thermal comfort was recorded through a guided interview and questionnaire survey during the measurement. The measured wind velocity ratio is highest in the long-linear high-rise building layout, and is the lowest in the mid-rise linear building layout. Eight-seven per cent of respondents felt fairly comfortable living in the long-linear high-rise building layout, only 7% less than the mid-rise building layout. ...........

Guo, W., Meng, X., Zhang, Y., Wang, N.

57199262830;57194581397;57194584688;57194582523;

Spatial development model of sustainable tourism town based on smart city

(2017) Agro Food Industry Hi-Tech, 28 (1), pp. 853-857.

https://www.scopus.com/inward/record.uri?eid=2-s2.0-85020975732&partnerID=40&md5=01cbd36b08a8a0f2ca20ec0ff56d3c2e

AFFILIATIONS: School of Economics and Management, Yanshan University, Institute of Modern Tourism Industry and Organization Development, Yanshan University, Qinhuangdao, Hebei, China;

School of Economics and Management, Yanshan University, Qinhuangdao, Hebei, China

ABSTRACT: With the strong support of the rapid development of Internet. "Intelligent" and "Wisdom" has become an important direction for the development of various industries. Smart city integrates the policy design and practice innovation in the three areas of urbanization, industrialization and information technology, which is the focus of China's new round of economic development. Therefore, the spatial development model of sustainable tourism town based on smart city was studied in this paper. On the basis of brief introduction of relevant theories and concepts, the sustainable development model is established based on practical demonstration to discuss the future spatial development and planning model of tourism towns. ..................

6. PLOS authors have the option to publish the peer review history of their article (what does this mean?). If published, this will include your full peer review and any attached files.

Reviewer #1: No

Reviewer #2: No

Reviewer #3: **Yes: **Dr. Grigorios L. Kyriakopoulos

---

## [Author Response · Author response to Decision Letter 0]

11 Oct 2022

Dear Editor，

We are truly grateful to yours and other reviewers’ critical comments and thoughtful suggestions on our manuscript(Basis of the Topological Structure Analysis of Streets and Lanes: A Case Study of the Xixing Ancient Town). 

Based on these comments and suggestions, we have made careful modifications on 

the original manuscript. All changes made to the text are in blue color. We hope the new manuscript will meet your journal’s standard. Below you will find our point-by-point 

responses to the reviewers’ comments/ questions. We hope that these revisions are 

satisfactory and that the revised version will be acceptable for publication in PLOS ONE journal. 

Thank you very much for your work concerning our paper. Wish you all the best!

Review comments for the manuscript: PONE-D-22-18649

Study on the Sustainable Renewal Strategy of Ancient Towns with Canals on the Basis of the Topological Structure Analysis of Streets and Lanes: A Case Study of the Xixing Ancient Town

The manuscript deals with an interesting research topic, having a well organized but poorly addressed research study. Therefore, there is room for significant methodological, morphological, and argumentation improvements to be made, prior it to be accepted for publication at the PLOS ONE journal. To this end the following review comments can be considered.

1) A rough estimation of chronological flash-back to the century of the Xixing ancient town’ foundation, it can be provided in the Abstract section (in approximate century B.C.).

Response: The historical background of Xixing ancient town has been replenished into the Abstract section.

2) In case of some input data or results derived from the analysis, then, a selected and representative portion of them they can be collectively denoted in Abstract section, enabling a more precise representation of the relevant analysis and findings in the Abstract extract, accordingly.

Response: In the Abstract section, the reflection and analysis of this study has been strengthened, and the innovation points have been refined and clarified.

3) At the Introduction section authors are recommended to reorganize the relevant theoretical coverage into two distinct and separated subsections, relevant to two subsections/subheadings of: a) Geographical overview of ancient towns in China and b) Chronological overview of ancients towns establishment and evolution to the today Chinese context. The spatial characteristics approached in section 2. Literature Review are well organized. However, a missing information of both narratives of sections Introduction, Literature Review is that there is absolutely missing numerical information-outcomes that could be collectively represented in the form of a Table at the end of section 2. There is not need authors to recall the narrative statements, but to enrich them with numerical information fitting to the already, or to-be-developed, subsections of main sections 1 and these data of 8-10 citations are adequate.

Response: We have conscientiously read the literature recommended by reviewers and have drawn on relevant content and methods, some of which have been discussed in the literature review and included in the Reference, and some have been adopted in the strategic perspective. At the same time, the relevant pictures are appended for the comparison between the past and the present in Xixing ancient town, and the research reflection is put forward for the existing problems.

4) At the Methodology section authors are recommended to briefly represent the land uses of the ancient town examined, in terms of: a) agricultural zone and area occupied (in km^2, or as percentage of the total area of the town), urban zones: pedestrian, streets and lanes (in km^2, or as percentage of the total area of the town), irrigation system and rainfall profile (in mm), domesticated area. 

Response: The quota of land use for buldings within the core protection area and the planning control area of Xixing ancient town have been added in Section 3.1 of the article.

5) In my opinion the heading/section of “Research Design” it is part of the Methodology section, thus, it can be transferred in it, accordingly. Besides, the main frame of headings should be only that of (adding those main headings, where missing): 1. Introduction, 2. Literature Review, 3. Methodology, 4. Results, 5. Discussion, 6. Conclusions and Future Research Orientations. Then, other main headings can be degraded to subheadings’ relocated into the aforementioned headings-frame. Moreover, starting from the 1. Introduction and end to 6. Conclusions and Future Research Orientations section, all embodied subheadings can be titled as: 1.1, 1.2, ……., 2.1, 2.2, …..3.1, 3.2, ….., 4.1, ….4.2, ……., in successive order and numbering, not to be plenary given.

Response: The structure of the article has been adjusted accordingly, especially in the Conclusions section.

6) At the Discussion section should be reorganized focusing on the today reflections and commercial commitments/services offered, in alignment with the stated research objective: “spatial data such as the number of commercial entities on each street segment, the area, area width, depth of single building and spatial syntax parameters is carried out”. To this end the following key-determinants can be briefly outlined: a) visiting the ancient towns in China, as an inseparable part of the today tourism industry, b) commercial, manufacturing and industrial activities and products/goods’ offered, c) the role of technology in the main community uses: agricultural sector, irrigation technologies, manufacturing and public services accommodated, d) role of governmental policies and motivations scheduled in supporting the ancient towns heritage, protection, and development of related to infrastructure at the nearby, to ancient towns, areas in China. To this end the “Further Discussion”:

Response: In the Discussion section, we adopted the constructive comments of the reviewers to include the above suggestions to the discussion part, highlighting the new discoveries and summaries of this study.

-It can be detached, “upgraded as main section”, and presented as separated “Discussion” section, since it contains purely argumentative statements, though all of them have to be accompanied by cross-citations that are currently missing.

-It can be accompanied by numerical information, data, quantitative results, enabling a more precise and comprehensive “guidance” of research outcomes, proposals, and recommendations to areas and contexts of the same research- and spatial- characteristics to be derived and applied.

7) Since Figures and all subfigures of 2 up to 6 are graphically liking to each other, it is highly recommended authors to either provide a more detailed information/interpretation of the comparing and contrasting them, revealing what are the distinct characteristics from the one subfigure to the other one. To this end authors can update these graphs by adding indicators of presentation in the form of arrows into these maps, it could be highly supportive and recommended.

Response: From Figure 2 to Figure 6, subfigures and description have been added and a chart comparison has been made in responding to the recommendation 8.

8) There is a long arrangement of numbers of 6 or more decimal digits and fluctuated precision, like that of lines 205, 217, 229-230, but also in Tables data. Therefore, revising and fixing the precision of all data to only 3 decimal digits it is highly recommended for the scopes of this manuscript.

Response: The parameters have been modified to 3 decimal places, and the charts have been added to facilitate identification and highlighting of the research focus.

9) Figure 7 can be presented in a white-coloured background, in better visual contrast with the trend-line dots to be achieved.

Response: Figure 7 has been modified.

10) Literature enrichment with more and relevant published papers it is recommended, thus, checking and citing among the below listed studies, where missing and where matching to the revised manuscript, it can be undertaken.

Response: Relevant literature has been supplemented and properly quoted in accordance with the recommendation of the reviewer.

Scopus

EXPORT DATE:12 Jul 2022

Wang, Z., Zhang, H., Yang, X., Li, G.

36469470000;57373042100;57373940300;57373581900;

Neighborhood streets as places of older adults’ active travel and social interaction – A study in Daokou ancient town

(2022) Journal of Transport and Health, 24, art. no. 101309, . Cited 1 time.

https://www.scopus.com/inward/record.uri?eid=2-s2.0-85121263625&doi=10.1016%2fj.jth.2021.101309&partnerID=40&md5=708e7f95f3c811844c51f5b25830e74f

DOI: 10.1016/j.jth.2021.101309

AFFILIATIONS: Department of Architecture, Henan University, Kaifeng, China; 

Department of Architecture, Henan University of Technology, Zhengzhou, China; 

Department of Architecture, Zhengzhou University, Zhengzhou, China

ABSTRACT: Introduction: Neighborhood streets are convenient places for older adults to engage in behaviors for active living, such as walking (active travel) and chatting with neighbors (social interaction). Street environments and older adults' active living in ancient towns need investigation. Taking Daokou ancient town in China as an example, this research observed older adults’ active travel and social interaction on two neighborhood streets and investigated the difference in social engagement between older-adult groups on different streets. Methods: On-site non-participant observation was conducted for four weekdays with seven 30-min sections per day. Data of 350 older adults’ active travel and social interaction on these streets were collected. Street environmental factors were measured and classified into four categories in terms of active-travel promotion: typology, motivators, functionality, and safety. To identify the differences in social engagement between the groups by street, one-way ANOVA tests were conducted after controlling for a significant confounding variable (daypart). Results: Among the older adults, the most popular type of active travel was independent walking (67%). Of their social interaction, the most popular types were staying and chatting (61%), group walking, and chess or card playing. On the street considered more age-friendly to active travel, older adults engaged in more social interaction in the mid-mornings and afternoons (p < 0.05). Conclusion: This study highlighted older adults' active living on neighborhood streets in ancient towns. The findings can be used to create street affordances for older adults’ active travel and social interaction, and produce healthy outcomes through the refinement of design and transportation policies and practice on street intervention. © 2021 Elsevier Ltd

AUTHOR KEYWORDS: Active living; Aging; Environment; Health; Historic

DOCUMENT TYPE: Article

PUBLICATION STAGE: Final

SOURCE: Scopus

Manakina, A., Nikolaeva, V.

56134235500;57218826705;

Formation of sustainable development of bicycle and pedestrian zones in a modern city

(2020) IOP Conference Series: Materials Science and Engineering, 890 (1), art. no. 012179, . 

https://www.scopus.com/inward/record.uri?eid=2-s2.0-85090348649&doi=10.1088%2f1757-899X%2f890%2f1%2f012179&partnerID=40&md5=c0bf49b51025943e083d8eaf6c93b826

DOI: 10.1088/1757-899X/890/1/012179

AFFILIATIONS: Industrial University, Tyumen, 625000, Russian Federation

ABSTRACT: The article is devoted to the current theme of bicycle and pedestrian space development in the developing city, as well as to attracting residents to the landscaped parts of the city for active recreation. The authors consider various types of activity, the implementation of which requires landscaped and safe areas, as well as enumerate the positive aspects associated with the introduction of active recreation in the daily life of residents, therefore, improve the quality of life and increase investment attractiveness of the region. The aim of the study is to substantiate the significance for the city of Tyumen of the creation and combination of the cycling and pedestrian zones, which shall contribute to the welfare in the sphere of the active recreation of citizens. The methods of the study are the following: analysis of the population dynamics in the city of Tyumen, which affects the socio-economic development of the region; analysis of the structure of the central part of the city of Tyumen, indicating the lack of greenery; consideration of the option of bicycle lane placement near the street Dzerzhinsky city of Tyumen. © Published under licence by IOP Publishing Ltd.

AUTHOR KEYWORDS: cycle paths; cycling; ecology; gardening; pedestrian traffic; transport infrastructure

DOCUMENT TYPE: Conference Paper

PUBLICATION STAGE: Final

SOURCE: Scopus

Yu, X., Ma, S., Cheng, K., Kyriakopoulos, G.L.

53865793000;57218326388;57218326647;6603382498;

An evaluation system for sustainable urban space development based in green urbanism principles-a case study based on the Qin-Ba mountain area in China

(2020) Sustainability (Switzerland), 12 (14), art. no. 5703, . Cited 24 times.

https://www.scopus.com/inward/record.uri?eid=2-s2.0-85088828018&doi=10.3390%2fsu12145703&partnerID=40&md5=f4c407b3b707d7b77b1350577b4e98e5

DOI: 10.3390/su12145703

AFFILIATIONS: School of Architecture, Chang'an University, Xi'an, 710061, China; 

Electric Power Division, Photometry Laboratory, School of Electrical and Computer Engineering, National Technical University of Athens, Athens, 15780, Greece

ABSTRACT: Since the 20th century, the deterioration of the ecological environment around the world has challenged urban space construction. With the development of urbanization, the consumption of resources and energy has increased, the level of biodiversity has decreased, environmental pollution is approaching the critical level, and the contradiction between human habitat activity and ecological environment has become increasingly prominent. The sustainable development of urban space along with its economic and social benefits, taking into account the quality of life and ecological environment, has become a new and important subject that needs to be explored. In this study, the indices of the evaluation system for sustainable urban spatial development in regions with underdeveloped economies but rich in ecological resources are arranged in sequence through the systematic coupling analysis of collaborative evaluation information and a quantitative analysis. The influences of urban space elements on sustainable urban development are disclosed. On the basis of the generated data, an evaluation system for sustainable urban spatial development with a complete set of information is proposed. The proposed system is applicable to urban spatial development evaluation in regions in China with underdeveloped economies but rich in ecological capital. First, the basic concept of system coupling is introduced, and a coupling relationship between urban sustainable development and urban space is proposed. Second, the elements of urban space and the sustainable development in the Qin-Ba mountain area are extracted, and the precedence diagram method is used to construct a sustainable evaluation system for urban space development in the Qin-Ba mountain area. Third, the sustainable evaluation process of urban spatial development is proposed. Finally, the sustainable evaluation system for urban spatial development in the Qin-Ba mountain area is applied to evaluate the urban spatial development in Shangluo, Qin-Ling Mountains, China. The results show that, among the investigated 14 indicators, the proportion of industrial land use mainly influences sustainable urban spatial development. As for the rest of the index factors, per capita green land area and green coverage ratio of built-up areas, per capita urban construction land area, proportion of forestry area, greening rate of built-up areas, total industrial dust emission density, proportion of cultivated area, and average volume fraction of residential areas are the secondary influencing factors of sustainable urban spatial development. The evaluation system in this research is constructed with the three aspects of "green coordination", "green development", and "green sustainability" of sustainable urban spatial development, and it complements the evaluation contents of urban-rural ecological space coordination, land resource protection, and green development community, and so on. The conclusion of this study not only can provide a useful reference for urban spatial development planning for underdeveloped ecological capital areas of China but also can provide a theoretical basis for the management and control policy of sustainable urban spatial development. © 2020 by the authors.

AUTHOR KEYWORDS: City spatial development; Green coordination; Green development; Green sustainability; Sustainable evaluation system

DOCUMENT TYPE: Article

PUBLICATION STAGE: Final

SOURCE: Scopus

Nomura, R., Shan, S., Mori, S.

57191923705;57204313367;37112638600;

Analyzing spatial structure of traditional houses in old towns with tourism development and its transformation toward sustainable development of residential environments in hexia old town, in China

(2018) Sustainability (Switzerland), 10 (10), art. no. 3809, . Cited 6 times.

https://www.scopus.com/inward/record.uri?eid=2-s2.0-85055187483&doi=10.3390%2fsu10103809&partnerID=40&md5=69acac765079e477ed50bff363e6ac4c

DOI: 10.3390/su10103809

AFFILIATIONS: Faculty of Engineering, Hokkaido University, Sapporo, 060-8628, Japan; 

Shanghai Urban Construction Design and Research Institute (Group) Co., Ltd., Shanghai, 200125, China

ABSTRACT: This study elucidates the spatial structure of traditional houses, and its transformation toward sustainable development of residential environments in old towns that are currently the focus of tourism development initiatives. Hexia old town in Huai'an District, China, was chosen for this study involving a field survey, questionnaire survey, and interviews. Data analysis identified three periods of residential transformation and three patterns of the residential transformation process. In all cases, there was low awareness of building conservation measures and lack of tourism development by the government, which has translated into ongoing residential transformations according to the demands of daily life. However, in the interests of tourism development, some businesses have started making use of vacant residences and traditional homes, and the relative proximity to work and home imply sustainability of residents' lifestyles under these conditions. An improvement plan for an old town requires a thorough understanding of the circumstances surrounding residential modifications executed by residents, the problems faced in the living environment, as well as efforts to increase the residents' awareness of the issue of sustainable living in that old town. © 2018 by the authors.

AUTHOR KEYWORDS: China; Living environment; Spatial structure; Tourism development; Traditional house

DOCUMENT TYPE: Article

PUBLICATION STAGE: Final

SOURCE: Scopus

Dičiunaite-Rauktiene, R., Gurskiene, V., Burinskiene, M., Maliene, V.

57201215996;8963424300;10040854900;24332653600;

The usage and perception of pedestrian zones in Lithuanian Cities: Multiple Criteria and comparative analysis

(2018) Sustainability (Switzerland), 10 (3), art. no. 818, . Cited 9 times.

https://www.scopus.com/inward/record.uri?eid=2-s2.0-85044007619&doi=10.3390%2fsu10030818&partnerID=40&md5=e9d368d976fc11b7515f76720018ed38

DOI: 10.3390/su10030818

AFFILIATIONS: Institute of Land Management and Geomatics, Faculty ofWater and Land Management, Aleksandras Stulginskis University, Studentu 11, Akademija, Kaunas distr., LT-53361, Lithuania; 

Road Department, Faculty of Environmental Engineering, Vilnius Gediminas Technical University, Sauletekio ave. 11, Vilnius, LT-10223, Lithuania; 

Department of the Built Environment, The Built Environment and Sustainable Technologies Research Institute, Faculty of Engineering and Technology, Liverpool John Moores University, Byrom street, Liverpool, L3 3AF, United Kingdom

ABSTRACT: As pedestrian zones are public spaces in urban areas, they are important in terms of meeting people's needs. However, it is worth noting that attention should be paid not only to the development of the physical infrastructure, but also to a sustainable balance between the socio-economic and environmental aspects. To guide urban planning and management initiatives towards more sustainable patterns, it is essential to re-examine the already existing characteristics of cities, establishing how they are used and perceived by inhabitants. The present research suggests environmental, economic and social criteria that determine a greater vitality of pedestrian zones in cities and better life quality for the inhabitants. A questionnaire survey was used to assess common attitudes regarding the research topic in major cities in Lithuania. The multiple criteria decision-making COPRAS (COmplex PRoportional ASsessment) method was used for the formation of a priority queue. The research results showed the attitudes of inhabitants towards pedestrian zones in Vilnius, Kaunas and Klaipeda. The inhabitants of these three cities regarded social and environmental criteria groups as the most significant. Contrary to what was expected and anticipated, respondents do not consider economic criteria as playing a key role in the sustainable preservation and development of pedestrian zones. © 2018 by the authors.

AUTHOR KEYWORDS: Assessment of pedestrian zones in Lithuania; Criteria for successful pedestrian zones; Multiple criteria analysis; Pedestrian zones in cities; Sustainability of pedestrian zones

DOCUMENT TYPE: Article

PUBLICATION STAGE: Final

SOURCE: Scopus

Yang, F.

36607241700;

Investigating wintertime pedestrian wind environment and user perception in dense residential neighbourhood in a city of hot-summer and cold-winter climate zone, China

(2017) Indoor and Built Environment, 26 (3), pp. 392-408. Cited 3 times.

https://www.scopus.com/inward/record.uri?eid=2-s2.0-85018181386&doi=10.1177%2f1420326X15620257&partnerID=40&md5=3afd66a42594b9f0e83e188c3d22fbdc

DOI: 10.1177/1420326X15620257

AFFILIATIONS: College of Architecture and Urban Planning (CAUP), Tongji University, 1239 Siping Road, Shanghai, 200092, China; 

Key Laboratory of Ecology and Energy-Saving Study of Dense Habitat, Tongji University, Ministry of Education, Shanghai, China

ABSTRACT: Urban wind environment could have an impact on pedestrian's comfort and safety, as well as pollution dispersion and building energy consumption. For cities in the hot-summer and cold-winter climate zone of China, a proper design residential neighbourhoods is important to facilitate urban ventilation in hot and transient seasons and to protect users from strong winds in cold season. This paper reports the results of field measurements and a questionnaire survey in a large residential development with three different types of housings. Micrometeorology measurement was carried out at the pedestrian height level as well as at a rooftop reference station. Pedestrians' subjective perception on wind and thermal comfort was recorded through a guided interview and questionnaire survey during the measurement. The measured wind velocity ratio is highest in the long-linear high-rise building layout, and is the lowest in the mid-rise linear building layout. Eight-seven per cent of respondents felt fairly comfortable living in the long-linear high-rise building layout, only 7% less than the mid-rise building layout. For similar housing forms in Shanghai, the wintertime wind shelter may not be critical compared with summertime ventilation requirement, and that the site planning and housing design should focus mainly on summertime wind channelling. © The Author(s) 2015.

AUTHOR KEYWORDS: Hot-summer cold-winter; Housing form; Pedestrian comfort; Urban design; Urban ventilation; Wind perception

DOCUMENT TYPE: Article

PUBLICATION STAGE: Final

SOURCE: Scopus

Guo, W., Meng, X., Zhang, Y., Wang, N.

57199262830;57194581397;57194584688;57194582523;

Spatial development model of sustainable tourism town based on smart city

(2017) Agro Food Industry Hi-Tech, 28 (1), pp. 853-857. 

https://www.scopus.com/inward/record.uri?eid=2-s2.0-85020975732&partnerID=40&md5=01cbd36b08a8a0f2ca20ec0ff56d3c2e

AFFILIATIONS: School of Economics and Management, Yanshan University, Institute of Modern Tourism Industry and Organization Development, Yanshan University, Qinhuangdao, Hebei, China; 

School of Economics and Management, Yanshan University, Qinhuangdao, Hebei, China

ABSTRACT: With the strong support of the rapid development of Internet. "Intelligent" and "Wisdom" has become an important direction for the development of various industries. Smart city integrates the policy design and practice innovation in the three areas of urbanization, industrialization and information technology, which is the focus of China's new round of economic development. Therefore, the spatial development model of sustainable tourism town based on smart city was studied in this paper. On the basis of brief introduction of relevant theories and concepts, the sustainable development model is established based on practical demonstration to discuss the future spatial development and planning model of tourism towns. The results show that the concept of sustainable development wisdom tourism towns is very necessary, and the establishment of spatial development model is the most important.

AUTHOR KEYWORDS: Smart city; Space development; Sustainable; Tourism town

DOCUMENT TYPE: Article

PUBLICATION STAGE: Final

SOURCE: Scopus

---

## [Decision Letter · Decision Letter 1]

20 Oct 2022

PONE-D-22-18649R1Study on the Sustainable Renewal Strategy of Ancient Towns with Canals on the Basis of the Topological Structure Analysis of Streets and Lanes: A Case Study of the Xixing Ancient TownPLOS ONE

Dear Dr. Wang,

Thank you for submitting your manuscript to PLOS ONE. After careful consideration, we feel that it has merit but does not fully meet PLOS ONE’s publication criteria as it currently stands. Therefore, we invite you to submit a revised version of the manuscript that addresses the points raised during the review process.

We look forward to receiving your revised manuscript.

Kind regards,

Dragan Pamucar

Academic Editor

PLOS ONE

Reviewers' comments:

Reviewer's Responses to Questions

**Comments to the Author**

1. If the authors have adequately addressed your comments raised in a previous round of review and you feel that this manuscript is now acceptable for publication, you may indicate that here to bypass the “Comments to the Author” section, enter your conflict of interest statement in the “Confidential to Editor” section, and submit your "Accept" recommendation.

Reviewer #1: (No Response)

Reviewer #3: All comments have been addressed

2. Is the manuscript technically sound, and do the data support the conclusions?

Reviewer #1: Partly

Reviewer #3: Yes

3. Has the statistical analysis been performed appropriately and rigorously? 

Reviewer #1: Yes

Reviewer #3: N/A

4. Have the authors made all data underlying the findings in their manuscript fully available?

Reviewer #1: Yes

Reviewer #3: Yes

5. Is the manuscript presented in an intelligible fashion and written in standard English?

Reviewer #1: Yes

Reviewer #3: Yes

6. Review Comments to the Author

Reviewer #1: This paper discusses the spatial syntax to analyze the topological structure of roads and alleys in the ancient city of Xixing in the Zhedong canal as a case object. I think this topic is interesting regard to most situations in urban development. But this paper should be improved in order to publishing requirement in the international journal. Specific comments are noted below:

1. Authors(s) already provided the abstract’s content (the purpose/background, method used, results). However, please provide clear methods on the studies that have been conducted. Please make it short and sharp. In addition, I recommend writing a clear statement about methodology. It is not necessary to write the steps of the method, but briefly explain the method used, analysis, and tools used. Write the keywords in alphabetical order.

2. In the initial paragraph in the introduction the problem is less clear from the author's point of view. It should refer after summarizing some of the results from the existing references. Some sentences that need to be corrected such as "...As a result, how to conduct innovative research on the space protection and vitality renewal of ancient towns with canals in the new historical period has become an important topic for the continuity and long-term development of those ancient towns in the modern era…” Further explanation is needed in this statement. The introduction is something that explains the contents of the entire manuscript. I suggest entering the contribution section in the introduction (contribution is part of introduction). Make the sentences flow and adjust the sentence by referring to the reference described in the initial paragraph in the introduction.

3. Did you mean Yoshinobu Ashihara? (line 77).

4. Literature review is a section that explains the basis of the study conducted. It is necessary to explain which parts of the reference are used in the methodology and then used to carry out the analysis. How the author(s) explains this into the methodology is important. It is highly recommended to summarize each part of the literature review by relating it to the research topic being carried out. State at the end of each paragraph each literature review (lines 90, 104, and 127).

5. Paper for international journals will be read by readers who do not understand the location, so give information in the form of maps of Xixing, China, and which parts of China are studied.

6. Explain with a statement at the end of the discussion section that explains the analysis of the discussion that has been described. Therefore, it is necessary to distinguish which part of the results and which part of the discussion. Make it in its own part.

7. Conclusion is the answer to the question asked in the introduction. So, the conclusion in this article should answer what is being questioned. Need to do a sharper explanation is actually what problems to be answered. In my opinion, there is no conclusion and it is too long for conclusion section, but it is also necessary to answer that should be at the end of the introduction. Please avoid writing conclusions in points, make in paragraph sentences. Two to three paragraphs for the conclusion section are sufficient.

8. Each figure requires a caption and explanation in the image, for example adding text to the captured image calculation from DepthMap.

9. In general, systematics in writing of scientific articles is Introduction, Literature review, Research Methods, Results and Discussion, and Conclusions. Systematics should be used in writing articles in this paper need to be improved.

This paper actually is an interesting topic to be raised; the author(s) tries to contribute the study regarding spatial syntax in the form of integration, choice, synergy, and clarity of streets and alleys, in order to obtain an update strategy and conclusions about the spatial vitality of a city. But this paper requires adjustment to the structure of a journal; it is necessary to follow the structure of the existing journal writing in general, so that the flow of writing becomes easy to understand.

Reviewer #3: At this revised manuscript authors developed a thorough and substantial reorganization of their initial manuscript, having the review comments addressed in a meticulous manner. The narrative flow is smooth, the methodology analysis is sound, and the research outcomes are insightful. In this respect the revised manuscript sustains novel features of regional and wider socio-cultural impact on the Chinese context, thus, it can be accepted for publication at the PLOS One journal as is.

7. PLOS authors have the option to publish the peer review history of their article (what does this mean?). If published, this will include your full peer review and any attached files.

Reviewer #1: No

Reviewer #3: **Yes: **Dr. Grigorios L. Kyriakopoulos

---

## [Author Response · Author response to Decision Letter 1]

11 Nov 2022

Dear Editor,

We are truly grateful to yours and other reviewers’ critical comments and thoughtful suggestions on our manuscript (Basis of the Topological Structure Analysis of Streets and Lanes: A Case Study of the Xixing Ancient Town). 

Based on these comments and suggestions, we have made careful modifications on the original manuscript. All changes made to the text are in yellow color. We hope the new manuscript will meet your journal’s standard. Below you will find our point-by-point responses to the reviewers’ comments/questions.

We hope that these revisions are satisfactory and that the revised version will be acceptable for publication in PLOS ONE journal. 

Thank you very much for your work concerning our paper. Wish you all the best!

Review comments for the manuscript: PONE-D-22-18649

Study on the Sustainable Renewal Strategy of Ancient Towns with Canals on the Basis of the Topological Structure Analysis of Streets and Lanes: A Case Study of the Xixing Ancient Town

Thank you for submitting your manuscript to PLOS ONE. After careful consideration, we feel that it has merit but does not fully meet PLOS ONE’s publication criteria as it currently stands. Therefore, we invite you to submit a revised version of the manuscript that addresses the points raised during the review process.

1. Authors(s) already provided the abstract’s content (the purpose/background, method used, results). However, please provide clear methods on the studies that have been conducted. Please make it short and sharp. In addition, I recommend writing a clear statement about methodology. It is not necessary to write the steps of the method, but briefly explain the method used, analysis, and tools used. Write the keywords in alphabetical order.

Response: The clear statement has been given, please see the new manuscript document.

2. In the initial paragraph in the introduction the problem is less clear from the author's point of view. It should refer after summarizing some of the results from the existing references. Some sentences that need to be corrected such as "...As a result, how to conduct innovative research on the space protection and vitality renewal of ancient towns with canals in the new historical period has become an important topic for the continuity and long-term development of those ancient towns in the modern era…” Further explanation is needed in this statement. The introduction is something that explains the contents of the entire manuscript. I suggest entering the contribution section in the introduction (contribution is part of introduction). Make the sentences flow and adjust the sentence by referring to the reference described in the initial paragraph in the introduction.

Response: The paragraph has been revised, please see line17-20.

3. Did you mean Yoshinobu Ashihara? (line 77).

Response: I mean in the ancient town, the original canal only providing water and aesthetics function, prefer to the limited function of original canal.

4. Literature review is a section that explains the basis of the study conducted. It is necessary to explain which parts of the reference are used in the methodology and then used to carry out the analysis. How the author(s) explains this into the methodology is important. It is highly recommended to summarize each part of the literature review by relating it to the research topic being carried out. State at the end of each paragraph each literature.

Response: This part has been revised, please see line90-91,104-105,125-127.

5. Paper for international journals will be read by readers who do not understand the location, so give information in the form of maps of Xixing, China, and which parts of China are studied.

Response: The new maps have been added, please see China map and Xixing ancient town location map.

6. Explain with a statement at the end of the discussion section that explains the analysis of the discussion that has been described. Therefore, it is necessary to distinguish which part of the results and which part of the discussion. Make it in its own part.

Response: The paragraph has been revised, distinguished the results, discussion and expectation of this study. Results was line 367-377.Discussion was 380-392. Expectation was 441-444.

7. Conclusion is the answer to the question asked in the introduction. So, the conclusion in this article should answer what is being questioned. Need to do a sharper explanation is actually what problems to be answered. In my opinion, there is no conclusion and it is too long for conclusion section, but it is also necessary to answer that should be at the end of the introduction. Please avoid writing conclusions in points, make in paragraph sentences. Two to three paragraphs for the conclusion section are sufficient.

Response: The structure of conclusions section has been adjusted, Please see line 428-432.

8. Each figure requires a caption and explanation in the image, for example adding text to the captured image calculation from DepthMap.

Response: All the figures in the manuscript have been advised ,please see the figure4-figure9.

9. In general, systematics in writing of scientific articles is Introduction, Literature review, Research Methods, Results and Discussion, and Conclusions. Systematics should be used in writing articles in this paper need to be improved.

Response: The systematics of the paper has been adjusted and improved accordingly.

---

## [Decision Letter · Decision Letter 2]

15 Nov 2022

PONE-D-22-18649R2Study on the Sustainable Renewal Strategy of Ancient Towns with Canals on the Basis of the Topological Structure Analysis of Streets and Lanes: A Case Study of the Xixing Ancient TownPLOS ONE

Dear Dr. Wang,

Thank you for submitting your manuscript to PLOS ONE. After careful consideration, we feel that it has merit but does not fully meet PLOS ONE’s publication criteria as it currently stands. Therefore, we invite you to submit a revised version of the manuscript that addresses the points raised during the review process.

We look forward to receiving your revised manuscript.

Kind regards,

Dragan Pamucar

Academic Editor

PLOS ONE

Journal Requirements:

Reviewers' comments:

Reviewer's Responses to Questions

**Comments to the Author**

1. If the authors have adequately addressed your comments raised in a previous round of review and you feel that this manuscript is now acceptable for publication, you may indicate that here to bypass the “Comments to the Author” section, enter your conflict of interest statement in the “Confidential to Editor” section, and submit your "Accept" recommendation.

Reviewer #1: (No Response)

Reviewer #3: (No Response)

2. Is the manuscript technically sound, and do the data support the conclusions?

Reviewer #1: Yes

Reviewer #3: Yes

3. Has the statistical analysis been performed appropriately and rigorously? 

Reviewer #1: Yes

Reviewer #3: N/A

4. Have the authors made all data underlying the findings in their manuscript fully available?

Reviewer #1: Yes

Reviewer #3: Yes

5. Is the manuscript presented in an intelligible fashion and written in standard English?

Reviewer #1: Yes

Reviewer #3: Yes

6. Review Comments to the Author

Reviewer #1: This paper discusses the spatial syntax to analyze the topological structure of roads and alleys in the ancient city of Xixing in the Zhedong canal as a case object. I think this topic is interesting regard to most situations in urban development. The author has revised most of what has been suggested. But there are minor revisions that must be corrected.

In this version of the article, the conclusion section is omitted. The author must write a conclusion, as stated in the previous revised version "Conclusion and Expectations", “…Based on the theory and method of spatial syntax, this paper studied the spatial and topological configuration…” In this version it is even removed. Stick to this conclusion statement. Please rewrite the section as in the previous version.

Reviewer #3: The following minor corrections can be addressed prior to an “acceptance” decision.

The manuscript title has to be shortened since it currently contains 31 words.

There are extensive narrative points that are not cited, thus, checking and cross-citing them it is needed.

It is not clear and straightforward what are the borders: start-end of the main sections: 1. Introduction, 2. Literature Review, 3. Methodology, 4. Results, 5. Discussion, 6. Conclusions and Future Research. Therefore, it is highly recommended:

-if main headings are missing to be added in the order above, while all main headings to be numbered as 1, 2, 3, …….., and their subheadings to be numbered as 2.2, 2.1, ……, 3.1, 3.2, ……., where missing and where matching to the narrative flow/text content.

-titles such as that of:

“Local integration degree of the Xixing Ancient Town”

“The choice of the Xixing Ancient Town”

“Construction of the Adaptive Mode of the Integration of Living Environment and Business Space”

To be shortened and placed as subheadings within the aforesaid main sections’ structure: 1,…., 6, where matching to the revised narrative flow.

The concluding remarks (to be presented in a separated 6. Conclusions and Future Research section) they have to oriented towards the constraints, the limitations, and the future research orientations and propositions of generalized truth, beyond that of Xixing ancient town, of similar research, cultural and spatial contexts in China or worldwide, to be drawn. To this end 3-4 extra sentences can be added at this ending section 6.

Special care and provision should be taken that all source citations from which the Figures and Tables have been developed/adapted/recalled from, to be explicitly noted in their legends/caption (where missing).

7. PLOS authors have the option to publish the peer review history of their article (what does this mean?). If published, this will include your full peer review and any attached files.

Reviewer #1: No

Reviewer #3: **Yes: **Dr. Grigorios L. Kyriakopoulos

---

## [Author Response · Author response to Decision Letter 2]

24 Nov 2022

Dear Editor,

We are truly grateful to yours and other reviewers’ comments on our manuscript (PONE-D-22-18649R2). 

Based on these comments, we have made careful modifications on the original manuscript. The revised parts according to the comments are marked in blue, and other changes are in yellow. Below you will find our point-by-point responses to the reviewers’ comments/questions.

We hope that these revisions are satisfactory and that the revised version will be acceptable for publication in PLOS ONE journal. 

Thank you very much for your work concerning our paper. Wish you all the best!

 Weilu Lv, Ning Wang, Yun Huang

Review comments for the revised manuscript: PONE-D-22-18649_R2

Study on the Sustainability of Ancient Canal Towns on the Basis of the Topological Structure Analysis of Streets and Lanes: A Case Study of the Xixing Ancient Town

The following minor corrections can be addressed prior to an “acceptance” decision.

The manuscript title has to be shortened since it currently contains 31 words. 

Response: We have already shorten the manuscript title. The new title is: Study on the Sustainability of Ancient Canal Towns on the Basis of the Topological Structure Analysis of Streets and Lanes: A Case Study of the Xixing Ancient Town.

There are extensive narrative points that are not cited, thus, checking and cross-citing them it is needed.

Response: We have checked the manuscript and cited the relevant points. Please see the new references.

It is not clear and straightforward what are the borders: start-end of the main sections: 1. Introduction, 2. Literature Review, 3. Methodology, 4. Results, 5. Discussion, 6. Conclusions and Future Research. Therefore, it is highly recommended:

-if main headings are missing to be added in the order above, while all main headings to be numbered as 1, 2, 3, …….., and their subheadings to be numbered as 2.2, 2.1, ……, 3.1, 3.2, ……., where missing and where matching to the narrative flow/text content.

-titles such as that of:

“Local integration degree of the Xixing Ancient Town”

“The choice of the Xixing Ancient Town”

“Construction of the Adaptive Mode of the Integration of Living Environment and Business Space”

To be shortened and placed as subheadings within the aforesaid main sections’ structure: 1,…., 6, where matching to the revised narrative flow.

Response: We have numbered the heading titles of the main sections. Please see the new manuscript document.

The concluding remarks (to be presented in a separated 6. Conclusions and Future Research section) they have to oriented towards the constraints, the limitations, and the future research orientations and propositions of generalized truth, beyond that of Xixing ancient town, of similar research, cultural and spatial contexts in China or worldwide, to be drawn. To this end 3-4 extra sentences can be added at this ending section 6.

Response: We have summarized the limitations and proposed two directions for future research to broaden the research perspective and deepen research thinking. Please see the 7. Expectations section.

Special care and provision should be taken that all source citations from which the Figures and Tables have been developed/adapted/recalled from, to be explicitly noted in their legends/caption (where missing).

Response: We have marked the sources of the Figures and Tables. Please see the new manuscript document.

---

## [Decision Letter · Decision Letter 3]

1 Dec 2022

PONE-D-22-18649R3Study on the Sustainability of Ancient Canal Towns on the Basis of the Topological Structure Analysis of Streets and Lanes: A Case Study of the Xixing Ancient TownPLOS ONE

Dear Dr. Wang,

Thank you for submitting your manuscript to PLOS ONE. After careful consideration, we feel that it has merit but does not fully meet PLOS ONE’s publication criteria as it currently stands. Therefore, we invite you to submit a revised version of the manuscript that addresses the points raised during the review process.

We look forward to receiving your revised manuscript.

Kind regards,

Dragan Pamucar

Academic Editor

PLOS ONE

Journal Requirements:

Reviewers' comments:

Reviewer's Responses to Questions

**Comments to the Author**

1. If the authors have adequately addressed your comments raised in a previous round of review and you feel that this manuscript is now acceptable for publication, you may indicate that here to bypass the “Comments to the Author” section, enter your conflict of interest statement in the “Confidential to Editor” section, and submit your "Accept" recommendation.

Reviewer #1: (No Response)

Reviewer #3: All comments have been addressed

2. Is the manuscript technically sound, and do the data support the conclusions?

Reviewer #1: Yes

Reviewer #3: (No Response)

3. Has the statistical analysis been performed appropriately and rigorously? 

Reviewer #1: Yes

Reviewer #3: (No Response)

4. Have the authors made all data underlying the findings in their manuscript fully available?

Reviewer #1: Yes

Reviewer #3: (No Response)

5. Is the manuscript presented in an intelligible fashion and written in standard English?

Reviewer #1: Yes

Reviewer #3: (No Response)

6. Review Comments to the Author

Reviewer #1: This paper discusses the spatial syntax to analyze the topological structure of roads and alleys in the ancient city of Xixing in the Zhedong canal as a case object. I think this topic is interesting regard to most situations in urban development. The author has revised most of what has been suggested. But there are minor revisions that must be corrected.

In this version of the article, the conclusion section is omitted. The author must write a conclusion, as stated in the previous revised version "Conclusion and Expectations", “…Based on the theory and method of spatial syntax, this paper studied the spatial and topological configuration…” In this version it is even removed. Stick to this conclusion statement. Please rewrite the section as in the previous version.

Reviewer #3: At this revised manuscript authors improved their analysis and findings into a satisfactory and meaningful manner. In this respect the revised manuscript sustains novel features and it can be accepted for publication at the PLOS ONE journal as is.

7. PLOS authors have the option to publish the peer review history of their article (what does this mean?). If published, this will include your full peer review and any attached files.

Reviewer #1: No

Reviewer #3: **Yes: **Dr. Grigorios L. Kyriakopoulos

---

## [Author Response · Author response to Decision Letter 3]

2 Dec 2022

Dear Editor,

We are truly grateful to the comments on our manuscript (PONE-D-22-18649R2). 

Based on the comments, we have made careful modifications on the original manuscript. The revised parts according to the comments are marked in blue. Below you will find our point-by-point responses to the reviewers’ comments/questions.

We hope that these revisions are satisfactory and that the revised version will be acceptable for publication in PLOS ONE journal. 

Thank you very much for your work concerning our paper. Wish you all the best!

 Weilu Lv, Ning Wang, Yun Huang

Review comments for the revised manuscript: PONE-D-22-18649_R2

Study on the Sustainability of Ancient Canal Towns on the Basis of the Topological Structure Analysis of Streets and Lanes: A Case Study of the Xixing Ancient Town

Reviewer #1: This paper discusses the spatial syntax to analyze the topological structure of roads and alleys in the ancient city of Xixing in the Zhedong canal as a case object. I think this topic is interesting regard to most situations in urban development. The author has revised most of what has been suggested. But there are minor revisions that must be corrected.

In this version of the article, the conclusion section is omitted. The author must write a conclusion, as stated in the previous revised version "Conclusion and Expectations", “…Based on the theory and method of spatial syntax, this paper studied the spatial and topological configuration…” In this version it is even removed. Stick to this conclusion statement. Please rewrite the section as in the previous version.

Response: We have already added the conclusion at the last section. Please see the 7. Conclusion and Expectations section.

Reviewer #3: At this revised manuscript authors improved their analysis and findings into a satisfactory and meaningful manner. In this respect the revised manuscript sustains novel features and it can be accepted for publication at the PLOS ONE journal as is.

Response: Thanks a lot!

---

## [Decision Letter · Decision Letter 4]

20 Dec 2022

Study on the Sustainability of Ancient Canal Towns on the Basis of the Topological Structure Analysis of Streets and Lanes: A Case Study of the Xixing Ancient Town

PONE-D-22-18649R4

Dear Dr. Wang,

We’re pleased to inform you that your manuscript has been judged scientifically suitable for publication and will be formally accepted for publication once it meets all outstanding technical requirements.

Kind regards,

Dragan Pamucar

Academic Editor

PLOS ONE

Additional Editor Comments (optional):

Reviewers' comments:

Reviewer's Responses to Questions

**Comments to the Author**

1. If the authors have adequately addressed your comments raised in a previous round of review and you feel that this manuscript is now acceptable for publication, you may indicate that here to bypass the “Comments to the Author” section, enter your conflict of interest statement in the “Confidential to Editor” section, and submit your "Accept" recommendation.

Reviewer #1: All comments have been addressed

2. Is the manuscript technically sound, and do the data support the conclusions?

Reviewer #1: Yes

3. Has the statistical analysis been performed appropriately and rigorously? 

Reviewer #1: Yes

4. Have the authors made all data underlying the findings in their manuscript fully available?

Reviewer #1: Yes

5. Is the manuscript presented in an intelligible fashion and written in standard English?

Reviewer #1: Yes

6. Review Comments to the Author

Reviewer #1: This paper discusses the spatial syntax to analyze the topological structure of roads and alleys in the ancient city of Xixing in the Zhedong canal as a case object. I think this topic is interesting regard to most situations in urban development. Author(s) have made improvements according to suggestions. With this version, I believe it can be published in this journal.

7. PLOS authors have the option to publish the peer review history of their article (what does this mean?). If published, this will include your full peer review and any attached files.

Reviewer #1: No

---

## [Editor Report · Acceptance letter]

27 Dec 2022

PONE-D-22-18649R4 

Study on the Sustainability of Ancient Canal Towns on the Basis of the Topological Structure Analysis of Streets and Lanes: A Case Study of the Xixing Ancient Town 

Dear Dr. Wang:

I'm pleased to inform you that your manuscript has been deemed suitable for publication in PLOS ONE. Congratulations! Your manuscript is now with our production department. 

Kind regards, 

on behalf of

Dr. Dragan Pamucar 

Academic Editor

PLOS ONE